# Insulin secretion deficits in a Prader-Willi syndrome β-cell model are associated with a concerted downregulation of multiple endoplasmic reticulum chaperones

Erik A. Koppes[1], Marie A. Johnson[1], James J. Moresco[2¤], Patrizia Luppi[3], Dale W. Lewis[4], Donna B. Stolz[3], Jolene K. Diedrich[2], John R. Yates, III[2], Ronald C. Wek[5], Simon C. Watkins[3], Susanne M. Gollin[4], Hyun Jung Park[4], Peter Drain[3], Robert D. Nicholls[1]*

1 Division of Genetic and Genomic Medicine, Department of Pediatrics, UPMC Children's Hospital of Pittsburgh, Pittsburgh, Pennsylvania, United States of America, 2 Department of Molecular Medicine and Neurobiology, The Scripps Research Institute, La Jolla, California, United States of America, 3 Department of Cell Biology, University of Pittsburgh School of Medicine, Pittsburgh, Pennsylvania, United States of America, 4 Department of Human Genetics, University of Pittsburgh School of Public Health, Pittsburgh, Pennsylvania, United States of America, 5 Department of Biochemistry and Molecular Biology, Indiana University School of Medicine, Indianapolis, Indiana, United States of America

¤ Current address: Center for the Genetics of Host Defense, UT Southwestern Medical Center, Dallas, Texas, United States of America
* robert.nicholls@chp.edu

**Data Availability Statement:** The RNA-seq FASTQ files and processed data associated from this study have been deposited in the NCBI Gene Expression

## Abstract

Prader-Willi syndrome (PWS) is a multisystem disorder with neurobehavioral, metabolic, and hormonal phenotypes, caused by loss of expression of a paternally-expressed imprinted gene cluster. Prior evidence from a PWS mouse model identified abnormal pancreatic islet development with retention of aged insulin and deficient insulin secretion. To determine the collective roles of PWS genes in β-cell biology, we used genome-editing to generate isogenic, clonal INS-1 insulinoma lines having 3.16 Mb deletions of the silent, maternal- (control) and active, paternal-allele (PWS). PWS β-cells demonstrated a significant cell autonomous reduction in basal and glucose-stimulated insulin secretion. Further, proteomic analyses revealed reduced levels of cellular and secreted hormones, including all insulin peptides and amylin, concomitant with reduction of at least ten endoplasmic reticulum (ER) chaperones, including GRP78 and GRP94. Critically, differentially expressed genes identified by whole transcriptome studies included reductions in levels of mRNAs encoding these secreted peptides and the group of ER chaperones. In contrast to the dosage compensation previously seen for ER chaperones in *Grp78* or *Grp94* gene knockouts or knockdown, compensation is precluded by the stress-independent deficiency of ER chaperones in PWS β-cells. Consistent with reduced ER chaperones levels, PWS INS-1 β-cells are more sensitive to ER stress, leading to earlier activation of all three arms of the unfolded protein response. Combined, the findings suggest that a chronic shortage of ER chaperones in PWS β-cells leads to a deficiency of protein folding and/or delay in ER transit of insulin and other cargo. In summary, our results illuminate the pathophysiological basis of

Omnibus (GEO) under SuperSeries GSE190337 including GSE190334 (total RNA-seq) and GSE190336 (small RNA-seq), assigned to BioProject PRJNA786769. The mass spectrometry (MS) proteomics data have been deposited to the ProteomeXchange Consortium via the PRIDE partner repository with the dataset identifiers PXD034471 (secreted peptides), PXD037874 (quantitative TMT MS; insoluble cellular proteins), and PXD037875 (soluble cellular proteins). A full set of bash scripts for analysis of both total and small RNA-seq data sets including custom annotations are provided at the github repository (https://github.com/KoppesEA/INS-1_PWS_RNA-Seq). All other numerical data and statistical analyses for graphical results reported in this manuscript are presented in S1 Data.

**Funding:** This work was supported by research grants to R.D.N. from the Eunice Kennedy Shriver National Institute of Child Health and Human Development (1R21HD108695) and from the Foundation for Prader-Willi Research (FPWR), and by funding to R.D.N. from the Storr Family Foundation through the Prader-Willi Syndrome Association (PWSA). J.J.M., J.K.D., and J.R.Y. were supported by the National Institute of General Medical Sciences (8P41 GM103533). The funders had no role in study design, data collection and analysis, decision to publish, or preparation of the manuscript.

**Competing interests:** The authors have declared that no competing interests exist.

pancreatic β-cell hormone deficits in PWS, with evolutionary implications for the multigenic PWS-domain, and indicate that PWS-imprinted genes coordinate concerted regulation of ER chaperone biosynthesis and β-cell secretory pathway function.

## Author summary

To investigate pancreatic islet beta-cell dysfunction in Prader-Willi syndrome (PWS), we engineered 3-megabase multigene deletions of the PWS imprinted domain in the INS-1 insulin-secreting cell line. The results establish that insulin secretion, critical for glycemic control, is impaired in PWS mutant beta-cells. Furthermore, genome-wide and focused RNA and protein studies in the PWS INS-1 model showed a stress-independent and chronic deficiency of multiple endoplasmic reticulum (ER) chaperones involved in the folding and subcellular transport of secretory pathway proteins, including hormones. Consequently, PWS beta-cells could not compensate levels for ER homeostasis and were more sensitive to ER stress activation. These results provide a mechanistic understanding of hormone deficits observed in PWS and implicate PWS-genes in coordinate regulation of ER chaperones and hormone secretion, affecting beta-cell secretory pathways that are dysregulated in common metabolic disorders.

## Introduction

Prader-Willi syndrome (PWS) is a multisystem disorder caused by loss of expression of a large contiguous cluster of paternally-expressed, imprinted genes from human chromosome 15q11.2 [1–3]. Clinically, PWS is characterized by failure to thrive with hypotonia, developmental and cognitive delay, behavioral problems, short stature, hypogonadism, hyperphagia, and early-onset obesity [3,4]. Prominent endocrine features of PWS include deficiencies of multiple hormones, including growth hormone, oxytocin, gonadotropins, insulin-like growth factor, thyroid hormones, amylin/IAPP, and pancreatic polypeptide [1,4–9]. In addition, plasma insulin is lower than expected in PWS relative to the degree of obesity [10–12]. Episodes of hypoglycemia have been reported in PWS patients, suggesting an imbalance in glucose homeostasis [13,14]. In contrast, plasma ghrelin is grossly elevated in PWS [3,10,15], perhaps as a physiological response to glucose imbalance [16,17]. Although the PWS literature suggests a hypothalamic etiology [3], the mechanisms for endocrine and metabolic dysfunction in PWS have not been elucidated.

The PWS-imprinted domain is comprised of ten paternally-expressed imprinted genes conserved in human and rodents (with an additional two genes each unique to human and rodents), encoding lncRNAs, snoRNAs, miRNAs, or distinct proteins [1,2,18–23]. These imprinted genes are predominantly expressed in neuronal [19,21,24] and neuroendocrine lineages [23], as well as pancreatic endocrine cells (α-, β-, δ-, and γ- cells, with only low expression in acinar, ductal, or undefined cells) based on recent gene-specific [25] and single cell RNA sequencing studies [26–28]. Further, *SNRPN* has decreased islet expression in β-cell failure due to saturated fatty acids [29] or cytokines [30], while *SNORD116* and *SNORD107* occur in islet exosomes and decrease by IL-1β and IFN-γ treatment [31]. Finally, significant reductions in β-cell gene expression occur for *SNORD116* in MODY3 [32] and for *SNURF* in type 1 diabetes [33]. These observations suggest that PWS-gene products function in the endocrine pancreas including in β-cells.

Our earlier studies of a transgenic-PWS (TgPWS) mouse model harboring a deletion of the orthologous PWS-imprinted domain demonstrated severe failure to thrive, abnormalities in fetal pancreatic islet development and architecture, reduced α- and β-cell mass, increased apoptosis, and postnatal onset of progressive hypoglycemia that led to lethality within the first postnatal week [16,25]. Plasma insulin and glucagon levels were low during fetal and neonatal life of TgPWS mice [16,25], and, at postnatal day 1 prior to onset of hypoglycemia, there was significantly reduced basal and glucose-stimulated insulin secretion (GSIS) from cultured TgPWS islets [25]. Furthermore, using an insulin-Timer fluorescent protein biomarker to image postnatal day 1 β-cells *in vivo*, TgPWS mice showed a striking accumulation of aged insulin whereas wildtype control littermates only displayed newly synthesized insulin [25]. These results suggested that PWS-imprinted genes are required for the development and secretory function of pancreatic endocrine cells.

Based on the salient insulin secretion dysfunction in the PWS-mouse model [25], we sought to assess the role of PWS genes in pancreatic β-cell secretory pathway function in a cellular model system. Herein, we used CRISPR/Cas9 genome editing within the INS-1(832/13) insulinoma (β)-cell line [34] to generate deletions of the complete PWS-domain. Following validation of a cell autonomous insulin secretion deficit in the PWS INS-1 model, we performed molecular profiling by proteomic and transcriptomic approaches. The data revealed deficits in multiple secreted hormones as well as endoplasmic reticulum (ER) chaperones that are components of secretory protein folding and trafficking pathways [35–40], providing mechanistic insight into PWS-gene function in pancreatic β-cells.

## Results

### Generation of INS-1 cell lines with silent, maternal allele PWS-gene deletions

Various insulinoma cell lines, predominantly of rodent origin, are widely utilized to investigate β-cell mechanisms as a tractable *in vitro* model system [41]. To investigate the functions of PWS-genes in β-cells, we used CRISPR/Cas9 genome-editing to target 3.16 Mb deletions encompassing the PWS-genes (**Figs 1A** and **S1A**) in rat INS-1(832/13)::mCherry insulinoma cells (hereafter termed INS-1) that secrete rat and human insulin [34] and express a mouse *Ins2* C-peptide-mCherry biosensor in insulin secretory granules [42]. The large deletions were visualized in about 9% of unselected, transfected cells as determined by fluorescence *in situ* hybridization (FISH) (**S1B Fig**). INS-1 lines with PWS-domain deletions were derived through sequential targeting and clonal isolation of cell lines initially harboring deletions of the silent maternal allele, followed by targeting of the remaining paternal loci, culminating in the creation of homozygous PWS-deletion lines (**Fig 1B**).

This process first led to isolation of two lines with a PWS-region deletion (5–5, 5–9) identified by deletion-breakpoint PCR (**Fig 1C**) and DNA sequencing (**S2A and S2B Fig**), and confirmed by FISH using BAC probes from within and outside the PWS-domain (**S1C–S1F Fig**). One PWS allele was deleted in all interphase and metaphase cells of line 5–5 (**S1D Fig**), indicating clonal isolation of a cell line with a PWS-domain deletion. Initially, line 5–9 was more complex, as FISH showed two cell types with either no deletion (most cells) or a PWS-domain deletion (**S1E Fig**); additionally, similar proportions of cells lacked mCherry fluorescence, likely due to epigenetic silencing (as DNA analysis indicated the transgene was present) or were mCherry-positive, respectively. This allowed separation by fluorescence-activated cell sorting (FACS), with establishment of a clonal mCherry-positive cell line (5–9) harboring a PWS-domain deletion (**S1F Fig**). Clonal engineered cell lines 5–5 and 5–9 were inferred to have maternal-deletions of the rat PWS-domain as evidenced by **1)** detection of only an

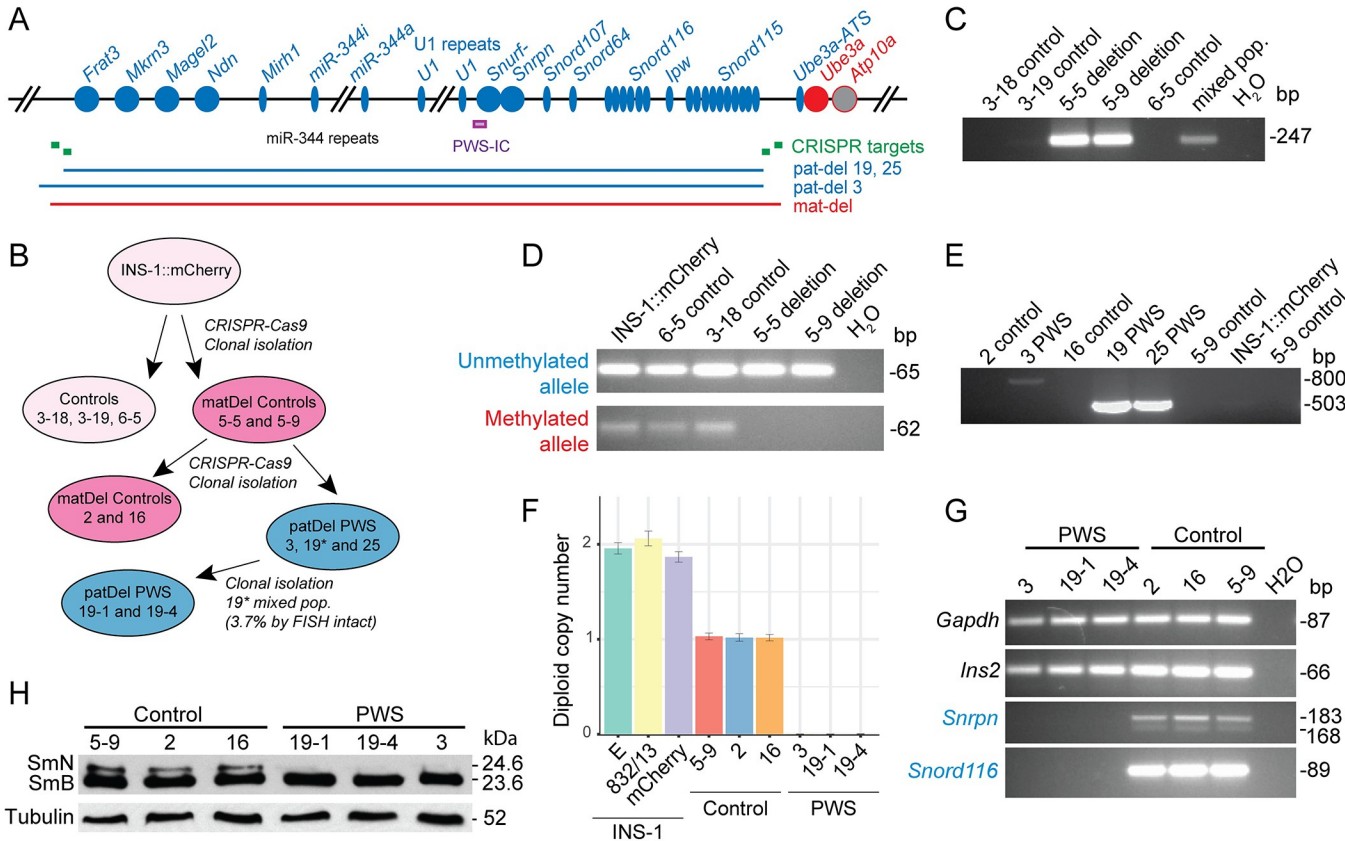

**Fig 1. CRISPR/Cas9 genome editing to generate INS-1 lines with 3.16 Mb deletions of the silent (maternal) and active (paternal) PWS-imprinted domain. (A)** Gene map of the rat PWS-imprinted domain and CRISPR/Cas9-targeted deletions. Symbols: circles, protein-coding genes; thin ovals, RNA genes; blue, (paternal; pat) and red (maternal; mat), imprinted genes; IC, imprinting control region (purple bar); green boxes, CRISPR gRNA target sites; blue and red horizontal bars, extent of deletions. Not all copies of tandemly repeated loci (*miR-344*; U1, *Snord116*, *Snord115*) are shown. **(B)** Schematic showing generation of genome edited, clonal INS-1 lines with 3.16 Mb deletions on the silent maternal allele only (5–9, 2, 16; dark pink) or homozygous deletion of both maternal and active paternal alleles (3, 19–1, 19–4, 25; blue). **(C)** First round deletion-PCR on genomic DNA identifies two INS-1 lines with deletions (5–9, 5–5). The deletion breakpoint is also detected in DNA from a heterogenous population of transfected cells (mixed pop.). **(D)** DNA methylation analysis of bisulfite-modified genomic DNA at the PWS-IC establishes that lines 5–5 and 5–9 have deletions on the maternal allele. **(E)** Second round deletion-PCR identifies three INS-1 lines (3, 19, 25) with deletions on the paternal allele, with line 3 having an alternate proximal breakpoint. **(F)** Gene dosage of *Snord107* normalized to *Ube3a* as determined by ddPCR using genomic DNA from the INS-1 panel of cell lines (also see **S9 Fig**). **(G)** PWS-deletion lines (3, 19–1, 19–4) lack mRNA expression of the PWS-imprinted genes, *Snrpn* and *Snord116*, as detected by RT-PCR, whereas control lines (5–9, 2, 16) express PWS-imprinted genes. All cell lines express the control genes (*Gapdh*, *Ins2*). The full panel of PWS-imprinted genes is shown in **S10A** and **S10B Fig (H)** Use of whole cell extracts for a western blot shows that PWS-deletion lines (3, 19–1, 19–4) lack expression of the spliceosomal SmN polypeptide (24.6 kDa) encoded by *Snrpn*, but retain expression of the paralogous SmB polypeptide (23.6 kDa) encoded by the unlinked *Snrpb* gene. The control is α-Tubulin.

unmethylated allele at the PWS-imprinting center (PWS-IC) by bisulfite PCR whereas parental INS-1 and control lines had both an unmethylated and a methylated allele (**Fig 1D**) indicative of biparental inheritance of the PWS-IC imprint; and **2)** expression of PWS-imprinted loci by reverse-transcription-PCR (RT-PCR) (**S1G Fig**), indicating the presence of an active, paternal-allele (**S1H Fig**). Finally, the presence of mutations at sgRNA target sites on the intact, paternal chromosome, termed scarred mutation alleles, identified a single nucleotide insertion at the sgRNA1 site (**S1I** and **S2C–S2E Figs**) but different nucleotide insertions at the sgRNA3 site (**S1I** and **S2F–S2H Figs**), the latter indicating an independent origin for the cell lines, 5–5 and 5–9. In contrast, no sequence changes occurred at top-ranked off-target sites for sgRNA1 or sgRNA3 (**S3 Fig**).

## Generation of PWS INS-1 cell lines with deletions on the active paternal allele

To specifically target the paternal allele of lines 5–5 and 5–9, we used rat-specific sgRNA sets internal to the first targeting sites to generate slightly smaller 3.143 Mb PWS-deletions (S4A–S4C Fig). The use of FISH demonstrated that about 5% of unselected, transfected cells had a PWS-deletion (S4D Fig). We chose maternal deletion (control) line 5–9 for a second round of transfections and clonal isolation, generating control lines 2 and 16 (Figs 1B, 1E, S4E and S4F) as well as PWS lines 19 and 25, with an expected paternal deletion-PCR product and PWS line 3 with a larger than expected deletion-PCR product (Figs 1B, 1E and S4G). DNA sequencing confirmed PWS-deletion breakpoints for lines 19 and 25 (S5A–S5C Fig) as well as for line 3, although the latter arose from a larger deletion at an alternate (alt) proximal alt-sgRNA70-3 targeting site with a DNA repair event that inserted a fragment of *E. coli* DNA in the breakpoint junction (S5D Fig). FISH studies confirmed that lines 3 and 25 had deletions of the PWS-domain on each allele (S4H and S4I Fig); however, FISH analysis also revealed that line 25 was mosaic for cells with diploid or tetraploid PWS-signals (S4I Fig). Consequently, line 25 was not used further. In contrast, line 19 had a residual fraction of cells (1–3.7%) with an intact paternal allele as shown by FISH (S4J Fig) and droplet digital PCR (ddPCR) (S6 Fig) which was removed by cell dilution and isolation of five clonal lines 19–1 through 19–5 (Figs 1B, S4K, S4L and S10A). Finally, sgRNA target sites not deleted or involved in a deletion breakpoint were assessed for scarred mutation alleles identifying a 28-nt deletion at alt-sgRNA70-3 in PWS lines 19–1 and 19–4 (S4N and S5E–S5L Figs), while no sequence changes occurred at top-ranked off-target sites for sgRNA70-3 (S7 Fig) or sgRNA79-1 (S8 Fig).

Genomic copy number was confirmed by ddPCR (Figs 1F, S6, and S9), establishing a set of three control lines (5–9, 2, 16) with hemizygosity for maternal-allele deletions (S1H Fig) and three PWS INS-1 lines (3, 19–1, 19–4) with homozygosity for PWS-domain deletions (S4M Fig). An absence of PWS-gene expression in all PWS lines was shown by RT-PCR for each PWS-imprinted gene (Figs 1G, S10A and S10B) and by western blot analysis for the SmN polypeptide encoded by *Snrpn* (Fig 1H). Intriguingly, PWS INS-1 lines showed a very low level of apparent expression of the PWS-imprinted gene *Snurf* by RT-PCR (S10A, S11A, and S11B Figs). Sequencing of the RT-PCR products identified an expressed *ψSnurf* sequence within a recently evolved *ψSnurf-ψSnrpn* locus located in an intron of the *Mon2* gene (S11B and S11C Fig). Using a *Pml*I variant between *Snurf* and *ψSnurf* (S11D Fig), we confirmed expression specifically of *ψSnurf* with complete silencing of *Snurf* (S11B Fig), as expected for PWS INS-1 lines. Scattered mutations in the *ψSnurf* (S11D Fig) and *ψSnrpn* (S11E and S11F Fig) segments likely inactivate any coding potential (S11G and S11H Fig). In summation, these results establish a panel of clonal INS-1 lines homozygous for ~ 3.16 Mb deletions, also termed PWS β-cell lines, and similarly, of clonal control lines with a deletion only involving the silent, maternal allele.

## PWS INS-1 cell lines show deficits in insulin secretion and ER chaperones

To determine whether PWS INS-1 cell lines have secretory deficits, we carried out insulin secretion assays under low (2.2 mM) and high (22 mM) glucose conditions. The three PWS cell lines displayed a striking deficit in both basal and glucose-induced insulin secretion as compared to the isogenic control INS-1 lines (Fig 2). However, both the PWS and control cell lines had a similar increase from each basal level for GSIS (Fig 2), indicating that PWS β-cells were not deficient in glucose responsiveness.

Next, mass spectrometry was used to assess changes in the cellular proteome and all secreted peptides in PWS *vs.* control INS-1 cell lines, under insulin secretion conditions of 22

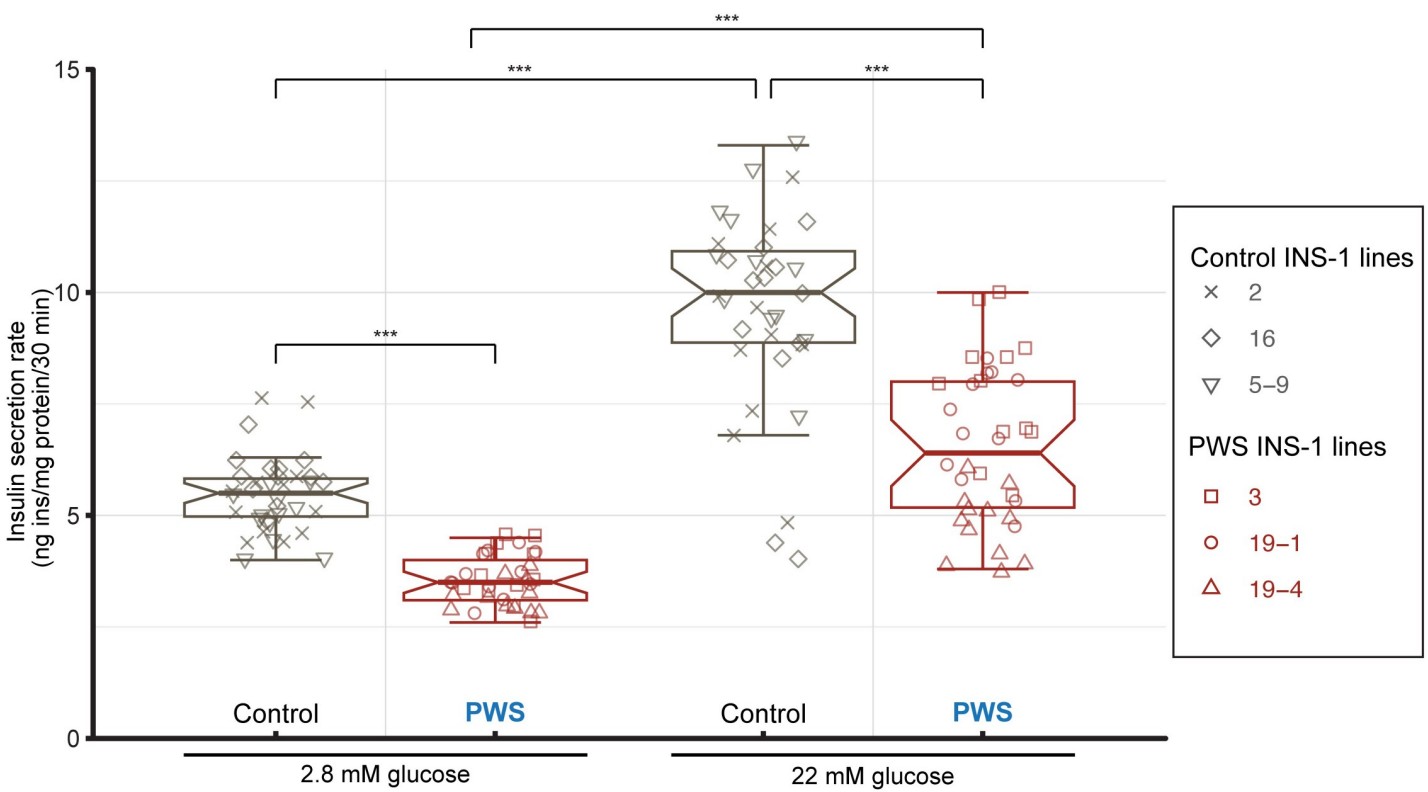

**Fig 2. Deficient basal and GSIS in PWS-deletion INS-1 lines.** Insulin secretory rates are presented as boxplots displaying the interquartile range (IQR) with notches marking the 95% confidence interval of the median and whiskers extending 1.5 x IQR for control (grey) and PWS (red) INS-1 lines at low (2.8 mM) and high (22 mM) glucose, with underlying data points indicated by unique symbols for each cell line (inset). Although GSIS increased 1.75-fold for control and 1.85-fold for PWS, loss of the PWS genes decreased the secretory rate by 36% in basal 2.8 mM glucose conditions and by 32% in stimulatory 22 mM glucose conditions. Statistical comparison (n = 36 biological replicates per group, with n = 12 per cell line) by ANOVA with Tukey's post hoc test (***, $P < 0.0001$).

mM glucose, separating cellular proteins by methanol-acetic acid extraction into soluble (mostly small) and insoluble (mostly large) protein fractions with the latter analyzed by quantitative proteomics (**Fig 3A**). Unexpectedly, in PWS INS-1 cells there was a striking deficiency of multiple ER chaperone proteins, including GRP78/BiP (HSPA5), GRP94/endoplasmin (HSP90B1), PDIA4, HYOU1, CRELD2, and DNAJB11, with lesser reductions in SDF2L1, DNAJC3, PDIA6, and PDIA3, and a modest decrease in PPIB (**Fig 3B**). Similar deficits of residual amounts of many of these ER chaperones as well as MANF were detected in the soluble protein fraction (**Fig 3C**). Furthermore, in PWS β-cell lines there was reduced abundance of numerous hormones co-secreted from INS-1 cells, including rat insulin-1 and insulin-2, mouse insulin-2, and human insulin (pre-pro-, pro-, C-peptide, and mature versions of each), as well as reductions in processed and precursor forms of IAPP and NPY, detected in both the cellular small protein fraction (**Fig 3C**) and residual amounts in the large protein fraction by quantitative proteomics (**Fig 3B**). In contrast, the full-length secretory granule protein, chromogranin B (CHGB), and the C-terminal CHGB (CCB)-peptide were increased 2-fold (**Fig 3B and 3C**). Extending both the insulin secretion data (**Fig 2**) and the cellular protein deficiencies (**Fig 3B and 3C**), mass spectrometry analysis of peptides secreted into the culture media demonstrated reduced levels of all forms of insulins and IAPP in PWS β-cell media compared to control (**Fig 3D**). There was also a reduction in secreted levels of the 14 amino acid WE14 peptide processed from chromogranin A (CHGA), but no changes in the CHGA precursor and

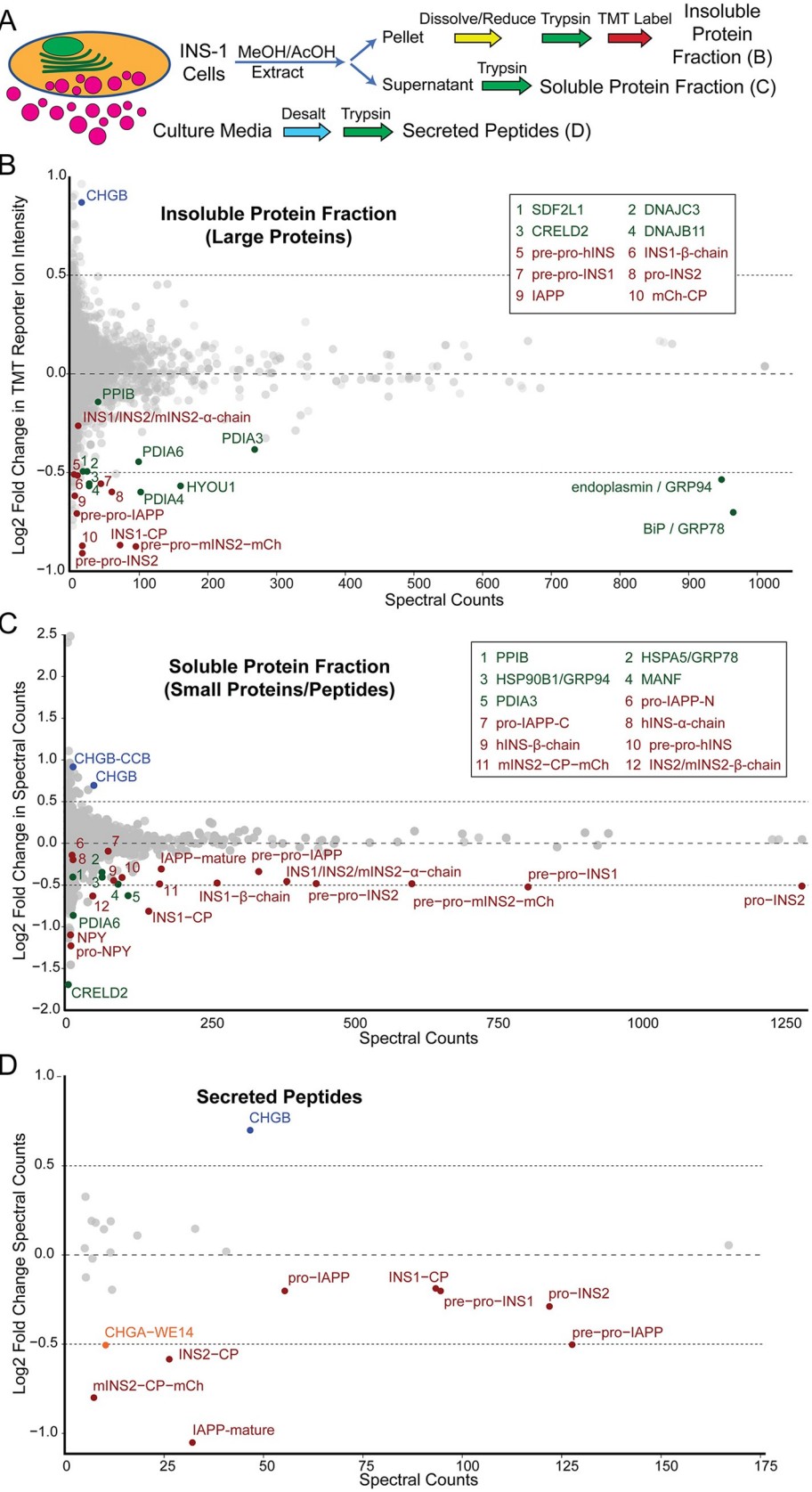

**Fig 3. Proteome-wide alterations in PWS-deletion *vs.* control INS-1 lines identifies reductions in levels of ER chaperones and hormones. (A)** Multiple fractions of PWS (3, 19–1, 19–4) and control (5–9, 2, 16) INS-1 cell cultures grown under GSIS conditions were assessed by mass spectrometry (MS). These included cellular proteins methanol-acetic acid extracted into an insoluble protein fraction assessed by quantitative Tandem Mass Tag (TMT) MS [see **Fig 3B**] or into a soluble protein fraction [see **Fig 3C**], as well as peptide hormones released from secretory granules (pink circles) into the media [see **Fig 3D**]. **(B-D)** Relative protein detection of proteins in PWS of Log2 Fold Change (PWS/Control) of protein detection plotted against spectral counts indicating overall protein abundance. **(B)** Relative comparison of insoluble cellular proteins in PWS and control INS-1 β-cell lines detected by quantitative TMT MS. Protein levels of eleven ER chaperones (green) as well as secreted hormones (red) insulin [various processed forms of INS1, INS2, mouse (m) mINS2-mCherry (mCh), human (h) INS, and C-peptide (CP); mature INS is the 21 amino acid peptide identical between rat INS1, rat INS2 and mouse INS2] and amylin (full length and processed IAPP) were markedly reduced in PWS INS-1 cell lines. In contrast, chromogranin B (CHGB) levels are noticeably increased in PWS INS-1 cell lines (blue). The boxed Key indicates the identity of the 10 numbered peptide spots. **(C)** Comparison of soluble cellular peptides detected in PWS *vs.* control INS-1 lines, with reductions observed in PWS lines for highly expressed peptide hormones (red) including insulins [various processed forms of INS1, INS2, mINS2-mCh and CP], amylin (processed and unprocessed IAPP), and neuropeptide Y (processed and unprocessed NPY), as well as lower levels in the soluble fraction of the ER chaperones (green). Additionally, two isoforms of chromogranin B (full-length CHGB and the CCB C-terminal fragment of chromogranin B) are increased in PWS INS-1 cell lines (blue). The boxed Key indicates the identity of the 12 numbered peptide spots. **(D)** Comparison of secreted peptides highlighting reduced levels for PWS relative to control INS-1 lines for numerous secreted hormones (red) including insulins [various processed forms of INS1, INS2, mINS2-mCh and CP] and amylin (processed and unprocessed IAPP), as well as reduced secreted levels from PWS INS-1 lines of the CHGA-derived WE14 peptide (orange), while secreted levels of CHGB (blue) were increased in PWS INS-1 cell lines.

processed forms in either secreted or cellular fractions. Secreted levels of chromogranin B were increased in PWS (**Fig 3D**), further illustrating the concordance between cellular and secreted peptide levels in PWS *vs.* control INS-1 lines. These results indicate that deletion of PWS-genes sharply lowers secretion of insulin and other peptide hormones (IAPP, NPY, CHGA-WE14) and this reduction is associated with deficiencies in many ER chaperones.

## Transcriptome studies of control and PWS INS-1 cell lines

Many of the PWS-imprinted genes (**Fig 1A**) are suggested to function in gene expression [18,19,21,22,43,44]. To gain insight into the molecular basis for insulin secretion and ER chaperone deficits in PWS INS-1 lines, high-throughput total RNA sequencing (RNA-seq) was used to identify differentially expressed genes (DEGs) under standard cell culture conditions (7.5 mM glucose) (**Figs 4A, 4B,** and **S12A and S1 and S2 Tables**). Visualization by a heatmap clustergram (**S12A Fig**) indicates that PWS clonal lines and control clonal lines each grouped together with similar expression profiles. A volcano plot depicting the magnitude, direction, and statistical significance of DEGs in the 3 PWS *vs.* 3 control lines illustrates no appreciable expression (Log2 Fold Change < -5) of PWS-imprinted genes with remaining DEGs symmetrically orientated around the ordinate axis with a near equivalent number of significantly upregulated (105) and downregulated (123) genes (**Fig 4A**). The PWS transcripts with complete loss of expression specifically in PWS INS-1 lines include *Snurf*, *Snrpn*, *Ipw*, *Mkrn3*, all four snoRNAs (*Snord116*, *Snord115*, *Snord107* and *Snord64*), miRNA (*Mir344* isoforms), and duplicated U1-*Snurf* sequences (**Fig 4A**), as also seen by RT-PCR (**Figs 1G, S10A, S10B, S11A and S11B**). Three PWS-imprinted genes, *Ndn*, *Magel2*, and *Frat3* (*Peg12*), were not detected as DEGs as these are not expressed by RT-PCR or RNA-sequencing in any of parental, control, or PWS INS-1 lines. These loci are present by genomic PCR, suggesting an epigenetic inactivation in the INS-1 founder cell line, although silencing is not widespread, as *Mkrn3* and *Mir-344* are interspersed with the silenced loci (see **Fig 1A**) and are well-expressed in control INS-1 lines (**S10B and S13 Figs**).

To ensure complete coverage of the INS-1 transcriptome, RNA-seq of small stable RNAs in the PWS *vs.* control cell lines was carried out, culminating in the identification of 58 significant

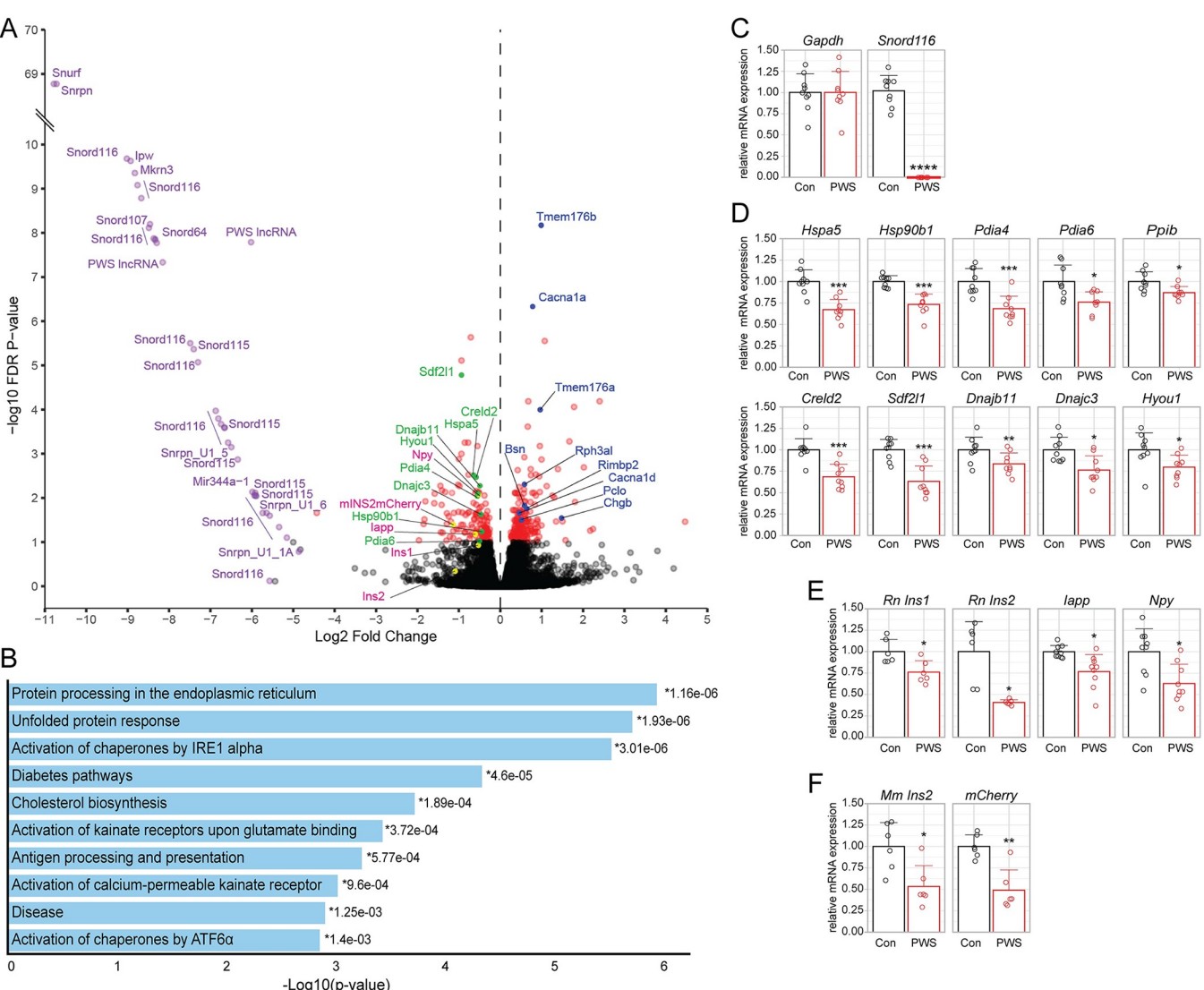

**Fig 4. Genome-wide transcriptome alterations in PWS-deletion *vs*. control INS-1 lines identifies significant differentially expressed genes (DEGs) including those encoding ER chaperones and hormones. (A)** Volcano plot showing statistical significance (-log10 FDR *P*-value) *vs*. magnitude of change (Log2 Fold Change) for all expressed genes. Colored data points indicate PWS-imprinted genes (purple), ER chaperones (green), secreted peptides (pink, with yellow dots), proteins involved in neuronal active zone/exocytosis and vesicle acidification (blue), and all other significant genes (red, without labels). **(B)** Gene set enrichment analysis identifies key biological pathways including those involved in protein processing and the unfolded stress response in the ER. **(C-F)** Quantitative gene expression analyses for control (Con, black; 5–9, 2, 16) *vs*. PWS (red; 3, 19–1, 19–4) determined by RT-ddPCR normalized to *Gpi* levels and to the average expression in control INS-1 lines, for **(C)** *Gapdh* and *Snord116* (PWS) control genes, **(D)** ten downregulated DEGs encoding ER chaperones, **(E)** endogenous *Rattus norvegicus* (Rn) genes encoding β-cell hormones, and **(F)** an exogenous transgene in the INS-1 lines encoding *Mus musculus* (Mm) *Ins2* with an mCherry biomarker. Statistical comparison by Welch's t-test: *, $P < 0.05$; **, $P < 0.005$; ***, $P < 0.0005$; ****, $P < 0.00005$.

differentially expressed miRNAs and snoRNAs (**S13A and S13B Fig** and **S3 Table**). Congruent with the RNA-seq data, all but three correspond to PWS-imprinted small RNAs, including duplicated *MiR-344*, *Snord116* and *Snord115* loci, as well as single copy *Snord107* and *Snord64* loci (**S13A and S13B Fig**). In addition, *Mir135b*, *Mir3065*, and *Mir212* were downregulated in PWS INS-1 lines (**S13B Fig**), although predicted targets were not DEGs by RNA-seq. Eleven of the top 25 highly expressed miRNAs in the rat INS-1 β-cell model (**S4 Table**), including *Mir375*, *Mir148a*, *Mir183*, *Mir30d*, *Mir27b*, *Mir25*, *Mir26a2*, *Mir26a*, *Mir192*, *Mir125a*, and *Mir141* are also in the top 25 expressed miRNAs for a human β-cell model [45]. Within INS-1

cells, expression of PWS-region *Snord116* copies make up 13.8% of the top 188 highly expressed snoRNAs and cumulatively would be the thirteenth highest expressed, with *Snord64* also in and *Snord107* just outside the top 100 (**S5 Table**), although snoRNAs are poorly studied as a small RNA class.

## Reduced mRNA levels for hormones and ER chaperones in PWS INS-1 cell lines

In addition to the loss of PWS-imprinted gene expression in the PWS INS-1 lines, four additional classes of DEGs were identified by manual annotation analysis of RNA-seq data (**Fig 4A**). These DEG classes encoded: **1)** hormones (e.g., *Iapp*, *Npy*, *mIns2::mCherry*) that were significantly reduced; **2)** nine ER chaperones that were lowered including *Sdf2l1*, *Hspa5*, *Creld2*, *Dnajb11*, *Hyou1*, *Pdia4*, *Dnajc2*, *Hsp90b1*, and *Pdia6*; **3)** "neuronal active zone" proteins that had increased expression, including *Cacna1a*, *Rph3al*, *Bsn*, *Rimbp2*, *Cacna1d*, *Pclo*, and *Chgb*, many of which also play a role in insulin exocytosis [46–51]; and **4)** two related transmembrane proteins involved in vesicle acidification (*Tmem176a,b*) [52] whose expression were also enhanced. Gene ontology and pathway analysis of downregulated genes (excluding PWS-imprinted genes) through Enrichr or DAVID highlighted enrichment of multiple ER functional groups including those in ER protein processing, and the unfolded protein response (UPR) that are linked with both IRE1 and ATF6 pathways (**Figs 4B** and **S14A–S14C**). Additional dysregulated pathways in PWS INS-1 lines included those relating to kainate receptors as well as antigen processing (**Fig 4B**). Finally, based on the input of downregulated DEGs, Enrichr predicted potential upstream transcriptional regulatory factors, including NFYA/NFYB (a cofactor of ATF6α, hereafter ATF6), CPEB1, RFX5, IRF3, CREB1, SREBF1, XBP1, and PPARB/D (**S15 Fig**).

Transcriptional changes observed by RNA-seq were corroborated by RT-droplet digital PCR (RT-ddPCR) analyses using RNA from independent biological replicates of each PWS and control INS-1 line (**Figs 4C–4F**, **S12B-S12**, **and S16E**). We validated equal expression of the housekeeping gene, *Gapdh*, and loss of expression of *Snord116* (**Fig 4C**), as well as significant downregulation in PWS INS-1 lines of 10 genes encoding ER chaperones (**Fig 4D**: *Hspa5*, *Hsp90b1*, *Pdia4*, *Pdia6*, *Ppib*, *Creld2*, *Sdf2l1*, *Dnajb11*, *Dnajc3*, and *Hyou1*), of endogenous hormones (**Fig 4E**: *Ins1*, *Ins2*, *Npy*, *Iapp*), and of the mouse *Ins2-mCherry* transgene (**Fig 4F**). Further, another ten downregulated genes in PWS β-cells were verified (**S12B Fig**) including *Syt1*, encoding a Ca²⁺-sensor [53] and *Jph3*, encoding a junctophilin involved in ER-plasma membrane contact bridges [54], each with roles in insulin secretion [53,54]; *Robo2*, encoding a receptor required *in vivo* for islet architecture [55], a process disrupted in TgPWS mice [25]; *Derl3*, involved in ER-associated degradation (ERAD) [56]; *Mylip*, encoding an E3 ubiquitin ligase regulating the LDL receptor [57]; *Atp10a*, encoding a lipid flippase [58]; *Tap1* and *Tap2*, encoding ER antigenic peptide transporters (**Fig 4B**) that play a role in type 1 diabetes [59]; and *Ndrg4* and *Id4*, of unknown relevance. Additionally, up-regulation of four genes was confirmed by RT-ddPCR (**S12C Fig**), including *Cacna1a*, *Chgb*, *Tmem176a*, and *Tmem176b*.

Several other observations merit note. There was a discrepancy for rat *Ins1* and *Ins2* mRNA levels between the RNA-seq (**Fig 4A**) and quantitative RT-ddPCR (**Fig 4E**) expression profiling, with only the latter showing statistically significant differences between PWS and control lines; despite a trend for reduced *Ins1* and *Ins2* in the PWS lines observed in the normalized RNA-seq data (**S16M Fig**). Only a minority of DEGs identified by RNA-seq did not validate by RT-ddPCR, including several encoding neuronal active zone proteins (**S12D Fig**). These minor discrepancies may be due to differences in cell culture environment or stochastic variability between the biological replicates. Additionally, an apparent increase in expression of

the human *INS-neoR* transgene in PWS lines (**S12C Fig**) was an artifactual consequence of epigenetic silencing of the transgene, specifically in control line 16 (**S16A, S16I–S16K and S16M Fig**). Interestingly, although both mRNA (*Pcsk1*) and protein levels of prohormone convertase PC1 were reduced in iPSC-derived neurons from PWS patients and in whole islets of inbred *Snord116*-deficient mice [23,60], neither PC1 protein nor *Pcsk1* or *Pcsk2* mRNA levels were changed in PWS INS-1 cell lines (**S12E Fig**). Finally, no markers of activation of apoptotic or other cell death pathways were observed by RNA-seq or proteomics, consistent with the absence of any increase in cell death observed in cultured PWS or control INS-1 cells. Indeed, electron microscopy showed normal mitochondria, rough ER, and other identifiable subcellular organelles in PWS and control INS-1 lines (**S17 Fig**). The difference in observations *in vivo* where increased apoptosis was observed in a subset of α- and β-cells in TgPWS fetal islets [25] and *in vitro* (this study) may reflect the use of enriched culture medium with reduced cellular stress under cell culture conditions. In summary, the transcriptome studies show that loss of PWS-gene expression in PWS β-cell lines is accompanied by alterations in levels of mRNAs encoding numerous secreted peptides and ER chaperones.

## Confirmation of insulin and ER chaperone protein deficits in PWS INS-1 lines

To further address the predicted disruptions of the ER and secretory pathway, we used western blot analyses to measure cellular levels of insulin and ER chaperones in control and PWS INS-1 lines. Levels of insulin polypeptides detected by an anti-insulin antibody were each significantly decreased in PWS INS-1 lines compared to controls, including diminished amounts of pro-insulin, pre-pro-insulin, and pro-mInsulin-2::mCherry bands (**Fig 5A and 5B**). Additionally, using an anti-mCherry antibody, we observed that PWS cells have significantly lower levels of the proinsulin form of the mouse insulin2-mCherry transgene but no decrease in the processed C-peptide (CP) form (**Fig 5C and 5D**). Importantly, use of a KDEL antibody to identify proteins with the ER retention motif confirmed significant deficiencies in levels of the two major ER chaperone proteins, GRP78/BiP and GRP94, in PWS INS-1 cell lines (**Fig 5E and 5F**). These results indicate that there are major disruptions of the protein folding machinery and attendant reductions in insulin processing and secretion in the PWS INS-1 cells.

## PWS INS-1 cells are more sensitive to activation by ER stressors

Chaperones, such as GRP78 and GRP94, not only have essential roles in protein folding and trafficking in the ER but disruptions of chaperone functions sensitize cells to agents that stress this organelle [61–66]. Therefore, we assessed whether the PWS INS-1 lines and their reduced levels of ER chaperones (**Figs 3–5**) would accentuate activation of the three sensory proteins of the UPR. First, ER stress activates IRE1α to catalyze mRNA processing of the *Xbp1* mRNA to generate the functional XBP1 transcription factor [66]; this non-canonical "splicing" of *Xbp1* transcripts occurs at low and equal levels in PWS and control INS-1 cells under DMSO control growth conditions but is enhanced by treatment with thapsigargin to initiate ER stress (**Figs 6A, 6B, S18A and S18B**). Importantly, production of "spliced" *Xbp1* mRNA in thapsigargin-treated cells occurs significantly more robustly at earlier 2 h and 3 h timepoints for PWS than for control INS-1 lines (**Figs 6A, 6B, S18A and S18B Fig**) but normalizes by 4–5 h (**S18A and S18B Fig**). These results reveal that while the initial magnitude of IRE1 activation is greater in the PWS INS-1 cells, the duration and cumulative response is comparable. Second, phosphorylation of eIF2α by PERK [65] was assessed, with PWS INS-1 lines showing significantly higher levels of eIF2α pSer51 phosphorylation than control lines when ER stress was induced by 5 h of thapsigargin treatment (**Fig 6C and 6D**). Phosphorylated eIF2α inhibits general translation

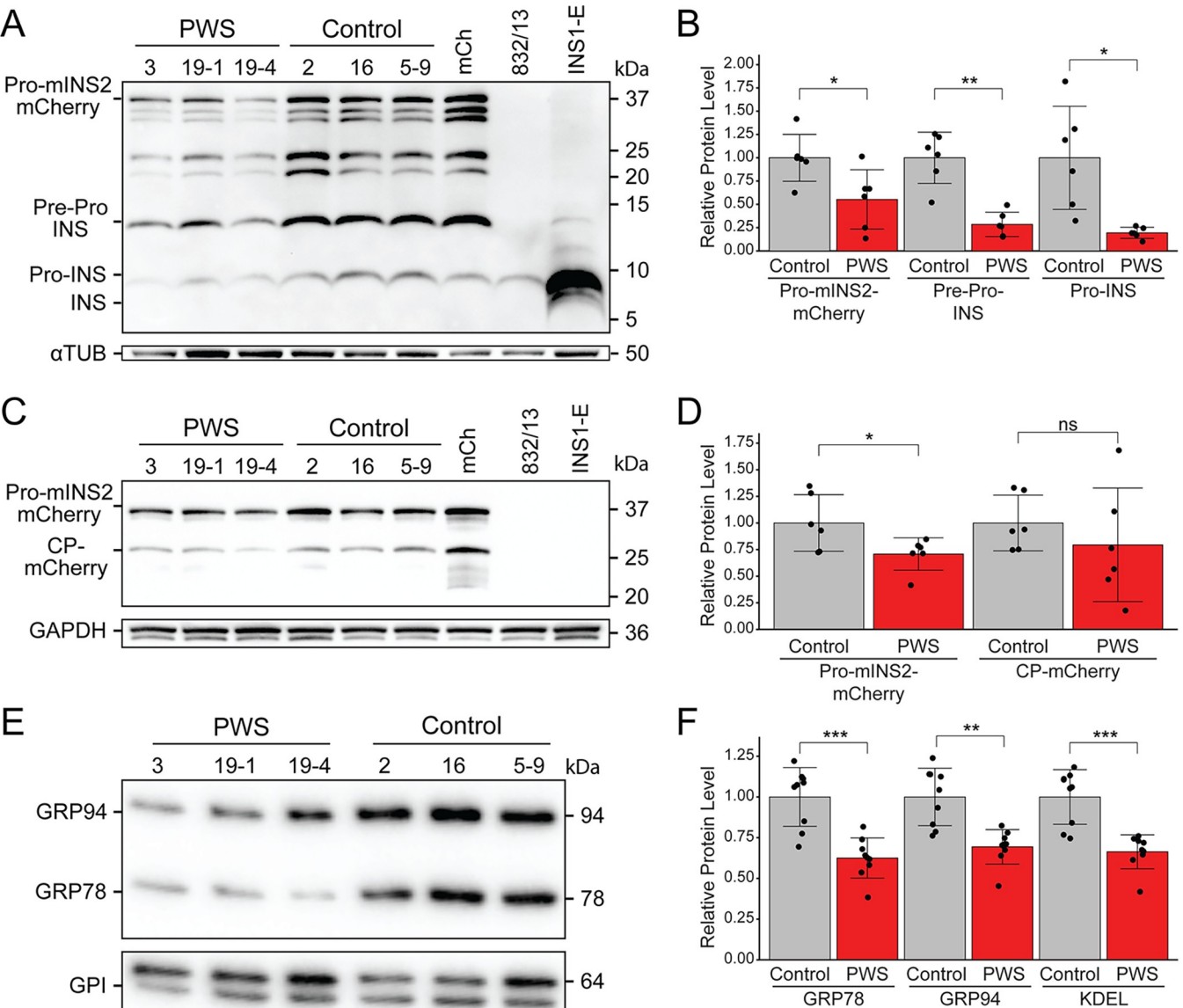

**Fig 5. Reductions in insulins and ER chaperone levels in PWS-deletion *vs.* control INS-1 lines.** Western blots of whole cell lysates from PWS (3, 19–1, 19–4) and control (5–9, 2, 16) INS-1 β-cell lines grown under control (DMSO) conditions for 5 h were assessed using a panel of antibodies. **(A)** Anti-insulin, detecting all cellular forms of insulin (Pre-Pro-, Pro-, and fully processed rat INS) as well as mouse proinsulin2 (Pro-mINS2)-mCherry. **(B)** Quantitation of Pro-mINS2-mCherry, Pre-Pro-INS and Pro-INS detected with anti-insulin in the PWS and control INS-1 lines (n = 6 each genotype). **(C)** Anti-mCherry, detecting mouse proinsulin2 (Pro-mINS2)-mCherry and C-peptide (CP)-mCherry. **(D)** Quantitation of Pro-mINS2-mCherry and CP-mCherry detected with anti-mCherry in the PWS and control INS-1 lines (n = 6 each). **(E)** Anti-KDEL, detecting the two major ER chaperones GRP94 (endoplasmin; HSP90B1) and GRP78 (BiP; HSPA5). **(F)** Quantitation of GRP78, GRP94, and total KDEL detected with anti-KDEL in the PWS and control INS-1 lines (n = 9 each). Anti-α-Tubulin, anti-GAPDH, and anti-GPI were used as controls for protein loading levels in **(A)**, **(C)**, and **(E)**, respectively. For **(A)** and **(C)**, control cell lines included INS-1::mCherry (mCh), INS-1(832/13) parental, and INS1-E. In **(A)**, unlabeled bands likely are Pro-mINS2-mCherry proteolytic fragments not containing the anti-mCherry epitope, as not detected in **(C)**. For **(B,D,F)**, statistical comparison by Welch's t-test: *, $P < 0.05$; **, $P < 0.005$; ***, $P < 0.0005$; ns, not significant.

but preferentially translates certain stress adaptive genes, including ATF4 and CHOP; transcripts of these genes were unaffected in unstressed PWS INS-1 cells as measured by RNA-seq.

Third, dissociation of ER-retained ATF6 from GRP78 by ER stressors such as de-glycosylation agents [62,63] enable it to move to the Golgi, where it is processed to an activated nuclear (N) form, ATF6-N, which regulates expression of genes encoding many ER chaperones,

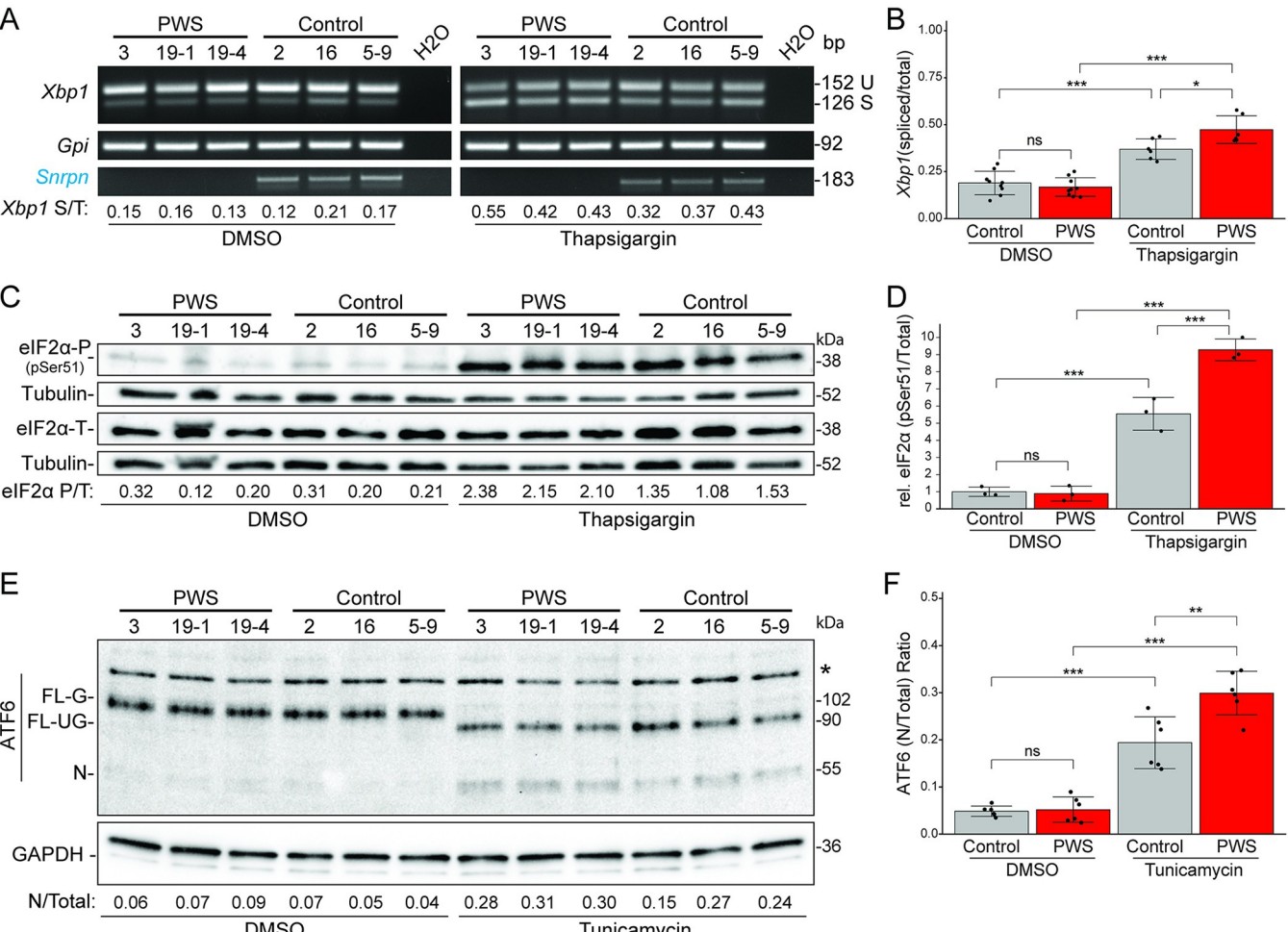

**Fig 6. More sensitive activation of all three ER stress master regulatory pathways in PWS-deletion *vs*. control INS-1 β-cell lines. (A)** For the IRE1α pathway, we assessed its mRNA processing of *Xbp1* from unspliced (U) to spliced (S), with control gene *Gpi* and PWS-*Snord116* also assayed by RT-PCR, using RNA from PWS (3, 19–1, 19–4) and control (5–9, 2, 16) INS-1 cell lines treated with DMSO or 0.1 μM thapsigargin for 3 h. At bottom, the *Xbp1* S/T (spliced/total) ratios for each cell line are shown, under both control (DMSO) and ER stress (thapsigargin) conditions. **(B)** Quantitation of the ratio of spliced/total (S/T) *Xbp1* mRNA in the PWS and control INS-1 lines under untreated (DMSO; n = 9 each genotype) and thapsigargin (3 h; n = 6 each) conditions. PWS β-cells are more sensitive to thapsigargin-induced ER stress with earlier and more robust activation of *Xbp1* mRNA "splicing" than control cell lines. **(C)** The PERK pathway was assessed by comparison of PERK phosphorylated eIF2α-P (pSer51) to total eIF2α, using whole cell lysates from PWS and control INS-1 cell lines treated with DMSO or 0.1 μM thapsigargin for 5 h. Anti-αTUB was used as a control for protein-loading levels. At bottom, the eIF2α P/T (phosphorylated/total) ratios for each cell line are shown under the untreated (DMSO) and ER stress (thapsigargin) conditions. **(D)** Quantitation of the relative (rel.) level of phosphorylated (pSer51) to total eIF2α in the PWS and control INS-1 lines (n = 3 each) under untreated (DMSO) and thapsigargin conditions. PWS cells are more sensitive to thapsigargin-induced ER stress with more robust activation of eIF2α phosphorylation. **(E)** For the ATF6 pathway, we used whole cell lysates from PWS and control INS-1 cell lines treated with DMSO or 10 μg/ml tunicamycin for 5 h. Under control DMSO conditions, full-length (FL) and glycosylated (G) ATF6 of ~ 100-kD is ER-localized. Tunicamycin inhibits biosynthesis of N-linked glycans, resulting in unglycosylated (UG) ATF6-FL of 90-kD in the ER, with proteolytic processing in the Golgi to produce nuclear (N) ATF6-N of ~ 55-kD. *, non-specific band detected by the anti-ATF6 antibody. Anti-GAPDH was used as a control for protein-loading levels. At bottom, the ATF6 N/Total (nuclear/Total) ratios for each cell line are shown under untreated (DMSO) and ER stress (tunicamycin) conditions. **(F)** Quantitation of the ratio of ATF6 nuclear/Total (N/Total) in the PWS and control INS-1 lines under untreated (DMSO) and ER stress (tunicamycin) conditions (n = 6 each, from two biological replicates). PWS cells are more sensitive to tunicamycin-induced ER stress with more robust activation of ATF6-N. For **(B,D,F)**, statistical comparison by ANOVA with Tukey's HSD post-hoc test: *, $P < 0.05$; **, $P < 0.005$; ***, $P < 0.0005$; ns, not significant.

including GRP78 and others. In PWS and control INS-1 cells, ER stress induced by tunicamycin, as expected, de-glycosylates full-length glycosylated (FL-G) ATF6 of ~ 102-kD to an unglycosylated 90-kD (FL-UG) form and the processed ATF6-N form of 55-kD (**Figs 6E** and **S19A**). The loss of the ATF6 FL-G form by deglycosylation occurs at a similar rate for PWS and

control cell lines (**S19B Fig**). However, the ratio of the processed ATF6-N (due to higher production in PWS than control INS-1 cells) to total ATF6 (**Fig 6F**) or of the ratio of ATF6-N to full-length unglycosylated 90-kD (with a higher level in control than PWS INS-1 cells) is significantly greater for PWS compared to control INS-1 cells at 2–5 h timepoints (**Figs 6E, S19A and S19B**). Thus, ATF6 is being activated to ATF6-N earlier and more robustly in PWS than in control INS-1 β-cell lines. Combined, these results on ER stress activation of the IRE1α/XBP1, PERK/eIF2α, and ATF6-N pathways indicate that PWS INS-1 β-cells are more sensitive than control INS-1 β-cells to ER stressors.

## Discussion

Although PWS has long been assumed to be a neuroendocrine disease of the hypothalamus [3,23], studies using mouse models have implicated a role for the pancreatic endocrine system [25,60]. Here, we generated a novel model through deletion of the ~ 3.16 Mb PWS-orthologous imprinted domain in INS-1 cells, as a proxy for β-cells, in which to investigate PWS β-cell biology. The PWS deletion results in a β-cell autonomous defect in production of peptide hormones and in deficiencies of β-cell ER chaperones required for folding, exit and trafficking of secretory peptides. Furthermore, the PWS deletion results in β-cell autonomous deficits in both basal and regulated insulin secretion, although the latter is at least in part consequent to the effect on basal insulin secretion, and a deficit in the secretion of other peptide hormones. Presciently, studies reported 25 years ago demonstrated that PWS adults compared to matched obese controls had significantly reduced first- and second-phase insulin secretion during intravenous glucose tolerance tests [11], attesting to the translational impact of the PWS INS-1 model. These findings strongly support a hypothesis that PWS-genes are required for fundamental β-cell mechanisms affecting production of peptide hormones.

### PWS β-cells have concerted deficits in ER chaperone expression and insulin production/secretion

Basal insulin secretion and GSIS are dramatically reduced both *in vivo* in the TgPWS-deletion mouse model [25] and the new *in vitro* PWS INS-1 β-cell model. As PWS β-cells in both models are glucose-responsive this indicates the defect is in a fundamental component of the secretory apparatus. In TgPWS mice, perinatal pancreatic insulin secretion dysfunction was associated with intracellular retention of aged insulin and reduced islet insulin and C-peptide content as well as reduced islet glucagon content and plasma glucagon levels suggestive of broader endocrine cell dysfunction [25]. Extending these observations, in our current study we have found that all insulin isoforms that are expressed in and secreted from the PWS INS-1 β-cell are decreased, as measured by proteomics and western blot analyses. Remarkably, these equivalent insulin secretion deficits between PWS mice [25] and the cell autonomous INS-1 model of PWS (this work) were associated with distinct transcriptional responses. *In vivo*, an apparent physiological compensatory mechanism increased levels of mRNAs encoding all major pancreatic hormones, including insulin, amylin, glucagon, somatostatin, and pancreatic polypeptide, and several secretory polypeptides although this failed to correct the secretory deficit [25]. In contrast, in PWS INS-1 cell lines with no neural or humoral input there was diminished mRNA expression of the endogenous rat *Ins1* and *Ins2* genes, a *mIns2::mCherry* transgene, and those encoding two other secreted peptides (*Iapp* and *Npy*). Taken together, the physiological regulation of insulins is dysregulated at multiple levels in PWS β-cells.

A further critical finding in the PWS INS-1 model is a concurrent down-regulation at the transcriptional and polypeptide levels of numerous ER chaperones, including for GRP78 and GRP94 that are major facilitators of insulin folding and trafficking in the ER. GRP94 directly

binds pro-insulin, and chemical inhibition or shRNA knockdown or genetic ablation of the *Hsp90b1* gene results in diminished insulin processing and secretion, larger but immature insulin granules, and activation of the PERK ER stress pathway [67]. Similarly, knockdown of *Hspa5* in INS-1(832/13) cells reduced insulin biosynthesis and secretion [68], a finding validated in human EndoC-βH1 cells [69]. The acute reduction of single ER chaperones resulting in reduced insulin production and secretion indicates a direct impact at the level of ER insulin protein folding and trafficking that is recapitulated in the PWS INS-1 model where we describe a chronic shortage of multiple ER-chaperones. Therefore, we interpret the broad ER-chaperone deficiency in the PWS INS-1 model as a primary molecular abnormality that contributes to diminished insulin secretion.

## Chronic stress-independent ER chaperone deficits in PWS β-cells prevents ER chaperone dosage compensation

Prior evidence from mouse knockout models of key ER chaperone genes has shown that they are subject to dosage compensation to maintain homeostasis. Although knockouts for either *Hspa5* (GRP78) or *Hsp90b1* (GRP94) are lethal early in mouse embryogenesis, heterozygous mice are viable [35,70–72]. Intriguingly, upregulation of GRP94 and other ER chaperones occurs in *Hspa5* +/- mice [70] and similarly for GRP78 in *Hsp90b1* -/- ES cells [71], indicative of dosage compensation among ER chaperones. A similar mechanism operates in *Hspa5* +/- mice to attenuate diet-induced obesity and insulin resistance [72]. Compensatory changes with upregulation of GRP94 or GRP78 and other ER chaperones including PDIA6 also occurred with shRNA knockdown of *Hspa5* or *Hsp90b1*, respectively, in a mouse cell line [73]. In contrast, the widespread deficiency of ER chaperones, including GRP78 and GRP94, that we describe for PWS INS-1 β-cells would preclude compensatory increases. Because of an inability to compensate, we hypothesize that PWS β-cells have a chronic deficit in ER chaperone production which would interfere with folding of proinsulin and/or delay ER transit of cargo (e.g., hormones) along the secretory pathway. Consistent with this hypothesis, our previous *in vivo* results using a Timer fluorescent protein demonstrated an accumulation of aged insulin in islets of PWS mice [25].

ER chaperones also function as sensors of ER stress, a physiological quality control mechanism responding to accumulation or failure in degradation of misfolded secretory proteins [3,61,64,74,75]. ER stress responses, governed by the IRE1α/XBP1, PERK/eIF2α/Ddit3 (CHOP), and ATF6 regulatory pathways, aim to recover ER homeostasis by upregulating ER chaperone gene expression as part of the UPR [39,61,64]. In unstressed cells, GRP78 forms stable complexes with each of IRE1α, PERK, and ATF6 in the ER lumen [61,62], whereas ER stress releases GRP78 and induces each pathway. Due to the demand to fold high levels of insulin, a protein notoriously difficult to fold [75], β-cells have a basal level of ATF6 activation to maintain higher levels of GRP78 than for most cell types [76]. In contrast, unresolved ER stress within β-cells leads to diabetes in mouse models and human [39,77]. For example, insulin gene mutations lead to insulin misfolding with induction of ER-stress and broad upregulation of the UPR [78], a converse mechanism to the reduced ER chaperone levels we demonstrated within PWS INS-1 cells under both basal conditions and during GSIS. Indeed, lower GRP78 levels in PWS β-cells likely accounts for the earlier and more robust ER stress induction by thapsigargin or tunicamycin by lowering the threshold required to dissociate ER stress activators resulting in cells that are poised to a greater degree for response by all three master regulatory pathways. Although normal β-cells *in vivo* cycle through states of elevated and low UPR coordinate with insulin gene expression and ER protein folding load [79–81], our omics data

on steady-state mRNA and protein levels of bulk PWS INS-1 cells under basal unstressed or GSIS conditions suggests a stress independent defect in ER chaperone gene expression.

## Mechanisms for dysregulated peptide hormone and ER chaperone expression in PWS β-cells

The coordinate downregulation of at least ten ER chaperone genes, including *Hspa5* and *Hsp90b1*, in PWS β-cells, but not the whole suite of UPR genes, supports the hypothesis that a PWS-imprinted gene or genes governs a pathway that coordinately regulates ER chaperones in a unique gene regulatory network (GRN). Similarly, the GRN may include genes encoding hormones given the corresponding coordinate reductions in expression. It is unknown whether the putative GRN is controlled via transcriptional, mRNA stability, and/or mRNA processing mechanisms. Although many of the DEGs that encode ER chaperones in PWS INS-1 β-cells are known ATF6 and/or XBP1 targets, numerous ATF6, IRE1α/XBP1, and ATF4/CHOP (PERK pathway) target genes [82–91] are not dysregulated in PWS INS-1 lines, suggesting a novel GRN in PWS β-cells. Candidate transcription factors (TFs) within the PWS-ER chaperone GRN are those with binding site motifs enriched in the promoters of DEGs in PWS β-cells, including, but not limited to the known ATF6 co-factor NFYA, which has roles in insulin secretion and glucose homeostasis [92], as well as PPARB/D also with known roles in β-cell mass and insulin secretion [93]. Intriguingly, a recently published study found that TFs of the nuclear receptor 4A (NR4A) family that participate in long-term memory in mice act through coordinate regulation of numerous ER chaperones [94], with a high degree of concordance with the dysregulated ER chaperones that we identified in PWS INS-1 cell lines; nevertheless, NR4A pathway factors are not DEGs in PWS β-cells.

An alternative hypothesis is that a reduction in secretory cargo (i.e., insulin) results in a lowered cellular requirement for ER chaperones. Insulin production has been successfully silenced at the post-transcriptional level using RNA interference in the mouse pancreas [95] or human insulinoma cells [69], but neither study assessed downstream cellular or metabolic consequences. In contrast, transcriptional and proteomic changes were assessed from islets of an insulin-deficient mouse model with tamoxifen-inducible deletion of *Ins2* alleles on an *Ins1* global knockout background (*Ins1 -/-*) [96]. At the single time point analyzed in this mouse model with ~ 50% reduced insulin production and prior to onset of diabetes and hyperglycemia, *Ins2* mRNA levels were 1.4-fold reduced (with no *Ins1* transcripts) and was associated with transcriptional and functional reductions in regulators of ER stress, including *Trib3*, *Atf5*, *Ddit3*, and *Xbp1* splicing, leading also to an increase in β-cell proliferation [96]. These reductions in insulin production are similar in level to the TgPWS mouse model [25] and to the PWS INS-1 model described here, nevertheless, in the tamoxifen-induced *Ins2* (*Ins1 -/-*) knockout mouse pancreas levels of mRNAs encoding ER chaperones were mostly unchanged (*Creld2*, *Pdia6*, *Dnajb11*, *Dnac3*, *Hyou1*, *Ppib*), to slightly increased (*Hsp90b1*, *Hspa5*) or decreased (*Sdf2l1*, *Pdia4*). These considerations therefore provide strong support for a GRN hypothesis for the coordinate reduction in ER chaperones in PWS β-cell models.

It remains to be determined which PWS gene or genes regulates the putative GRN. Neither *Magel2* nor *Ndn* have a role, as they are not expressed in INS-1 cells. The top candidate PWS genes that are highly expressed in INS-1 cells are *Snrpn*, encoding the SmN spliceosomal protein [43], *Snurf*, encoding a small arginine-rich nuclear protein [18], and *Snord116*, a tandemly duplicated C/D-box snoRNA that may function as a sno-lncRNA that sequestrates FOX2 and other RNA binding proteins in alternative splicing pathways [22,97], or through direct binding and regulation of as yet undefined RNA targets. Further studies are necessary to distinguish

amongst the candidate PWS genes and to identify downstream GRN steps leading to coordinated ER chaperone expression.

## Evolution of PWS gene functions in secretory endocrine cells

Recent work has shown hypothalamic deficiencies in *Magel2*-null mice as well as for iPSC- and dental-pulp stem cell neuronal models, specifically in secretory granule components including PCSK1, PCSK2, CPE, granins (e.g., CHGB and others), and numerous neuropeptides [23]. These studies established that MAGEL2 plays a role in hypothalamic neuroendocrine cells by regulating neuropeptide production and secretion via endosome recycling to prevent lysosomal degradation of secretory granule proteins [23], a process downstream of the ER in protein trafficking and secretory pathways. As discussed above, our study shows that a different PWS-gene (e.g., *Snrpn*, *Snurf*, and/or *Snord116*) is putatively responsible for regulating ER chaperone and hormone (e.g., insulin) biosynthesis and secretion from pancreatic β-cells. In contrast to the reported reduction of prohormone convertase PC1 in PWS iPSC-derived neurons and in *Snord116*-null mice [60], no deficiency in *Pcsk1* mRNA or PC1/3 levels or in *Pcsk2* or PC2 was observed in PWS β-cell lines, which may reflect differences in cell type, genetics, or experimental conditions. Combined, these observations in hypothalamic neuroendocrine cells and pancreatic β-cells indicate that at least two PWS-imprinted genes regulate related neuropeptide and peptide hormone secretory pathways, suggesting that the PWS domain may function as a "mammalian operon" or synexpression group [98,99]. As the PWS-domain arose evolutionarily in a eutherian mammalian ancestor [100], this suggests emergence of functions acting as molecular rheostats to regulate secretory pathways in endocrine cells that control growth, metabolism, and neural pathways.

## PWS genes regulate glucose and hormone homeostasis: implications for metabolic disease

Maintaining glucose homeostasis relies on exquisite coordination between secretion of the opposing hormones insulin and glucagon from pancreatic β- and α-cells, respectively, in response to changes in blood glucose, and disruption of these processes contributes to the pathogenesis of metabolic disorders, including type 2 diabetes [39,101–104]. Although insulin and glucagon play opposite regulatory roles both in normal metabolic homeostasis and dysfunction in type 2 diabetes, PWS mice are remarkable in having low blood levels of both insulin and glucagon [16,25]. As recent studies indicate that GRP78 interacts with glucagon in α-cells [105], it will be important to further assess dysregulation of ER chaperones and glucagon secretion in PWS α-cells both in cell culture and within PWS mouse models. Intriguingly, quantitative trait locus (QTL) studies in mice link genetic variation in blood insulin and glucose levels in or near to the PWS-orthologous domain [106]. Further studies are warranted to examine the role of PWS-imprinted genes in these metabolic traits in humans, and whether β-cell or other endocrine cell deficits in ER chaperones and hormone secretion contribute to clinical phenotypes, including episodes of hypoglycemia in PWS subjects [13,14]. Indeed, recent studies in mice show that hypoglycemia induces hunger and food intake acting through brainstem neurons projecting to the hypothalamus [107], and it will be important to assess the potential contribution of such mechanisms to hyperphagia and obesity in PWS. As PWS genes regulate islet development, β- and α-cell secretion [16,25], and a GRN affecting ER chaperone and insulin secretion (this study), identifying mechanisms by which PWS-genes carry out these critical β-cell functions will illuminate the pathogenesis and may reveal effective treatments for not only PWS, but for common disorders with deficits in glycemic homeostasis and islet hormone secretory pathways.

## Materials and methods

### Cell culture

INS-1 lines were cultured using "RPMI 1640 media without glucose" (Life Technologies) supplemented with 10% FBS, 10 mM HEPES, 1 mM sodium pyruvate, 0.05 mM 2-mercaptoethanol, 7.5 mM glucose, and antibiotics (1% Pen-Strep, 10 μg/ml each piperacillin and ciprofloxacin). Chemical treatments included 0.1 μM thapsigargin (Sigma) for either 3 h, 5 h or a 0–5 h time-course, and 10 μg/ml tunicamycin (Sigma) for 5 h or a 0–8 h time-course. The INS-1 lines used for genome editing in this study were generated from parental INS-1(832/13) cells that have integrated a human *INS*-neoR transgene [34] as well as a mouse *Ins2*-C-mCherry transgene [42].

### Genome editing

Pairs of sgRNAs (**S6 Table**) were designed (http://crispr.mit.edu/) to target sites flanking the rodent PWS-orthologous domain to delete all paternally-expressed imprinted genes (**Fig 1A**), and cloned into the pX330 CRISPR/Cas9 vector [108] (Addgene). INS-1 parental cells were transfected using lipofectamine 3000 with pX330-vectors encoding sgRNAs targeting proximal of *Frat3* and distal (between *Snord115* and *Ube3a*) of the PWS-domain and/or the control pEGFP-N3 vector (using 3 μg or 500 ng of DNA per vector per T75 flask or 6-well plate, respectively). After 4 days culture at 30˚C or 37˚C, cells were harvested for DNA isolation or flow cytometry (FACS ARIAII in the Rangos Flow Cytometry Core Laboratory) with ~ 300 GFP-positive cells plated in five 96-well plates and clonally expanded to 12-well plates for DNA isolation. Clonal lines were screened by deletion-PCR and positive lines further expanded for DNA, RNA, protein, and molecular cytogenetic analyses. PCR primers for deletion-PCR, inversion-PCR, and scarred- or intact-allele PCRs are in **S7 Table**. For DNA sequencing (Genewiz) of deletion breakpoints and scarred or intact sgRNA sites, we Sanger sequenced PCR products directly or from pJET (Thermo Fisher) cloned PCR products. Potential CRISPR/Cas9 off-target sites were predicted *in silico* using CRISPOR (http://crispor.tefor.net/); top predicted off-targets were PCR amplified (**S7 Table**) and directly Sanger sequenced.

### Molecular cytogenetics

Cytogenetic studies were carried out in the University of Pittsburgh Cell Culture and Cytogenetics Facility. Briefly, cultured INS-1 cell lines were treated with 0.1 μg/ml Colcemid 1 h, harvested, fixed, and slides processed for metaphase FISH by standard methods [109]. Fluorescent probes were prepared by labeling BAC (BACPAC Genomics) DNA using nick translation (Enzo Life Sciences, Inc.), including CH230-114P11 from the central PWS-domain (encodes U1A-*Snurf-Snrpn-Snord107-Snord64*) with Orange-dUTP and control CH230-2B12 mapping several Mb distal of the PWS domain (encodes *Cyfip1-Nipa2-Nipa1-Herc2*) with Green-dUTP. Probe and slide preparation, hybridization, and DAPI staining were by standard cytogenetics methods, with FISH analyses on an Olympus BX61 epifluorescence microscope (Olympus Microscopes) and image capture and analysis using the Genus software platform on the Cytovision System (Leica Microsystems) [109].

### Droplet digital PCR (ddPCR) genomic copy number assays

For genomic copy number 5 ng of *EcoR*I digested genomic DNA was used as input with amplification of *Snord107*, PWS-IC, and *Mirh1* as targets and *Ube3a* as a reference; during clonal derivatization EvaGreen chemistry was used whereas Fam/Hex TaqMan probes were used in a final analysis. Absolute copy number of *Snord107* and Poisson confidence intervals were

calculated by numerical approximation [110]. Primers and probes for copy number ddPCR are listed in **S7 Table**.

### DNA methylation

Genomic DNA was bisulfite converted using the EZ DNA methylation Gold kit (Zymo Research). Outer first round genomic PCR was performed using primers to amplify the *Snurf-Snrpn* promoter region (annealing temperature or Ta of 64˚C; **S7 Table**), with subsequent second round genomic PCR performed using PCR primers specific for the maternal, methylated or paternal, unmethylated alleles (Ta of 60˚C; **S7 Table**). HotStart Taq polymerase (Qiagen) was used for all methylation PCR.

### Insulin secretion assays

Cells were plated at $1.0 \times 10^6$ cells/well, then 24 h later at ~ 80% confluency were washed with PBS containing $Mg^{++}$ and $Ca^{++}$ (Gibco), pre-incubated 1.0 h in KRBH (129 mM NaCl, 5 mM $NaHCO_3$, 4.8 mM KCl, 1.2 mM KH2PO4, 1.2 mM $MgSO_4$, 2.5 mM $CaCl_2$, 10 mM HEPES, 0.1% BSA) with low glucose (LG; 2.8 mM) and washed in KRBH (without glucose). Following a 30 min incubation in KRBH-LG, 1.0 ml low glucose secreted fractions were removed and centrifuged 13,000 rpm for 5 min., aliquoted and stored at -80˚C. The cells were then incubated 30 min in KRBH-high glucose (HG; 22 mM), and high glucose secreted fractions similarly centrifuged and stored at -80˚C. Finally, cells were washed in PBS containing $Mg^{++}$ and $Ca^{++}$, harvested in RIPA lysis buffer and protease inhibitor, centrifuged at 13,000 rpm for 15 min at 4˚C and the supernatant (protein) stored at -80˚C. Protein was determined using Pierce BCA protein assay kit (Thermo Fisher). Insulin was measured by the RIA and Biomarkers Core of the Penn Diabetes Research Center, University of Pennsylvania, using an ultrasensitive rat insulin ELISA kit (Alpco Diagnostics). ANOVA followed by the Tukey's HSD post-hoc test was used to assess differences between conditions and genotypes using Prism 8 (GraphPad Software, Inc).

### Reverse transcription-PCR

Total RNA was isolated from PWS and control INS-1 lines grown as above (7.5 mM glucose) by Trizol harvest and miRNeasy (Qiagen) column purification. RNA quality was assessed by RNA TapeStation (Agilent) and quantified by broad range Qubit (Thermo Fisher) fluorometric analysis. First strand cDNA synthesis from 1 μg RNA was carried out using random hexamer primed RT by using Super Script IV (Thermo Fisher). Primers for gene specific RT-PCR amplification are in **S8 Table**. Quantification of the ratio of "spliced" *Xbp1* isoform to total *Xbp1* was determined by ethidium bromide gel densitometry with Image Lab (Bio-Rad).

### RT-Droplet digital-PCR (RT-ddPCR) assays

Template cDNA pools were generated from 1 μg RNA using cells grown under standard glucose conditions (as above) by using iScript (Bio-Rad) RT with first-strand synthesis priming from a mixture of random hexamers and oligo-dT. Droplet generation, PCR and reading of singleplex EvaGreen-based ddPCR reactions was carried out using an automated QX200 ddPCR system (Bio-Rad). A total template input volume of 2 μl of cDNA diluted in a range from undiluted to 1:400 (40 ng to 0.10 ng RNA equivalents per reaction) determined empirically based on the absolute expression level of target genes was used per 20 μl reaction. Primer sequences are listed in **S8 Table**. Expression of target genes was normalized to the stable and modestly expressed *Gpi* (1:5 diluted, 8.0 ng RNA equivalents). Relative expression of each

target gene was calculated using multiple independent RNA biological samples and technical replicates, with removal of rare statistical outliers greater than two standard deviations, by normalizing to both *Gpi* as a reference gene, and the average expression level in control lines. Comparison of relative target gene expression in PWS and control lines was evaluated using the Welch's t-test for unequal variance with a significant threshold of $P < 0.05$.

## Proteomics

Each of the 3 control cell lines (5–9, 2, 16) and the 3 PWS cell lines (3, 19–1, 19–4) were grown in 6-well plates, harvested and then washed with PBS containing $Mg^{++}$ and $Ca^{++}$, incubated 30 min in KRH-high glucose (22 mM, glucose-induced insulin secretion condition, with BSA excluded from the KRBH buffer) after which the media was collected, centrifuged 10 min 1300 rpm, and supernatant (secretory fractions 1–6; **Fig 3A**) collected and stored at -80°C. Cellular proteins were harvested by direct addition of 1.0 ml of methanol-acetic acid (90% methanol, 9% water, 1% acetic acid) to each well, transferred to a microcentrifuge tube and centrifuged at 1300 rpm 15 min. The supernatant (soluble protein fractions 7–12; **Fig 3A**) and pellets (insoluble protein fractions 13–18; **Fig 3A**) were collected and stored at -80°C. The secretory and cellular fractions were dried *in vacuo* and stored at -80°C. In preparation for mass spectrometry, proteins from each fraction were dissolved in 8 M urea, 100 μM Triethylammonium bicarbonate (TEAB) pH 8.5, reduced with tris(2-carboxyethyl)phosphine (TCEP) and alkylated with chloroacetamide, then diluted to 2 M urea with 100 mM TEAB, addition of 0.5 μg trypsin (Promega) and placed in a 37°C shaker for 16 h. Secretory protein fractions were analyzed on an Orbitrap Elite mass spectrometer (Thermo Fisher) while cellular protein fractions were analyzed on an Orbitrap Fusion mass spectrometer (Thermo Fisher). The 6-plex Tandem Mass Tag (TMT) system for quantitative proteomics (Thermo Fisher) was used to compare methanol-acetic acid insoluble cellular proteins (fractions 13–18) for control and PWS cell lines. The TMT labeled samples were analyzed on an Orbitrap Fusion Lumos mass spectrometer (Thermo Fisher).

Peptide/protein identification and quantification were determined using IP2 (Integrated Proteomics Applications). The MS raw data files were converted using RawConverter [111] (version 1.1.0.23) with monoisotopic option. For peptide identification, tandem mass spectra were searched against a database including the UniProt Rat database one entry per gene (21589 entries released 5/30/2021), these entries scrambled between K and R to make a decoy database, common contaminants, peptides and custom proteins using ProLuCID [112], and data filtered using DTASelect [113]. Quantitation of TMT samples was calculated with Census version 2.51 [114] and filtered with an intensity value of 5000 and isobaric purity value of 0.6 [115]. The Quantitative@COMPARE feature of IP2 was used to determine statistical significance.

## Immunoblot analyses

Cells were grown under standard glucose conditions with or without the addition of DMSO, thapsigargin or tunicamycin (as above). Whole cell lysates were harvested by direct lysis in cold radioimmunoprecipitation buffer (RIPA) with the addition of EDTA and combined protease and phosphatase inhibitors (Thermo Fisher) followed by clearing of insoluble material by centrifugation at 13,000 RPM for 10 minutes. Proteins were separated by SDS-PAGE with 2-mercaptoethanol as a reducing agent using criterion sized 4–15% gradient tris-glycine for broad range of molecular weights greater than 30 kDa or 16.5% tris-tricine for high resolution of smaller proteins (Bio-Rad). Proteins were transferred to nitrocellulose membranes and probed with antibodies [commercially obtained except for ATF6 [65], with dilutions as in **S9**

Table]. Chemiluminescence detection of HRP-conjugated secondary antibodies was performed with either Clarity or Clarity Max Western ECL detection kits on a Bio-Rad Chemi-Doc XRS+ imager. Densitometry measurements were made using Image Lab (Bio-Rad) software and exported for analysis in R. Protein normalization to a reference protein (i.e., GPI, GAPDH, or α-Tubulin) was made by dividing the band intensity for the target protein by the reference intensity and then subsequently by the average expression of the target protein in control cell lines. Statistical comparison of normalized protein levels between PWS and control samples was made by either Welch's unequal variance t-test or by ANOVA with Tukey's HSD post-hoc test for multiple comparison where appropriate, with a significance threshold of $P < 0.05$.

## Transmission electron microscopy

Monolayers of INS-1 cells were fixed in 2.5% glutaraldehyde in 100 mM PBS, post-fixed in aqueous 1% osmium tetroxide, 1% $Fe_6CN_3$ for 1 h, and dehydrated prior to embedding in Polybed 812 resin (Polysciences). Ultrathin cross sections (60 nm) of the cells were obtained on a Riechart Ultracut E microtome, post-stained in 4% uranyl acetate for 10 min and 1% lead citrate for 7 min. Sections were viewed on a JEOL JEM 1400 FLASH transmission electron microscope (JEOL) at 80 KV. Images were taken using a bottom mount AMT digital camera (Advanced Microscopy Techniques).

## RNA-seq and bioinformatics

Total RNA (from cells grown under 7.5 mM glucose) for each of the 3 control cell lines (5–9, 2, 16) and the 3 PWS cell lines (3, 19–1, 19–4) was used to prepare rRNA depleted Illumina TruSeq 75-bp paired-end stranded (fr-firststrand) sequencing libraries and sequenced to a depth of ~ 40 million reads on a NextSeq500 instrument at the UPMC Children's Hospital of Pittsburgh Genomics Core. RNA-seq data analysis was performed using computational resources available from the University of Pittsburgh Center for Research Computing. Quality of reads was assessed by FastQC and sequencing adapters trimmed by cutadapt using the combined TrimGalore tool [116]. The sequencing reads were aligned to the Ensembl v98 rat genome to which the sequence and annotation of the human *INS*-neo[R] and mouse *Ins2*-C-mCherry transgenes had been added along with annotation of previously undefined genomic features (e.g., *Ipw*, *ψSnurf*, and upstream *U1 Snurf-Snrpn* exons) using the STAR splice-aware aligner [117]. Gene feature counts were tabulated with HTSeq using either the default (—non-unique none) or customized (—nonunique all) options to enable the inclusion of multi-copy (i.e., *Snord116*) and overlapping (bicistronic *Snurf-Snrpn*) genes which otherwise were excluded as ambiguous reads under the default options [118]. Differential expression analysis was performed with DESeq2 using a cutoff of *P*adj < 0.1 as calculated by the Benjamini-Hochberg multiple comparison procedure [119]. Similar overall transcriptomic results were obtained with RSEM based analysis [120] but had lower counts for both multi-mapped reads for *Snord116* and for ambiguous bicistronic *Snurf-Snrpn* reads. Gene ontology enrichment analysis of up- and down-regulated gene sets was done using DAVID [121]. Additional gene ontology and upstream analysis utilized Enrichr [122].

Sequencing libraries of small RNAs including snoRNAs were made from the same total RNA isolation but using a miRNA-Seq (Qiagen) library prep kit with a modified size selection for up to 200-bp fragments. Single end reads were generated and sequenced. UMI tools [123] and Cutadapt [124] were used to deduplicate and remove adapters in preprocessing of the fastq reads for alignment with Bowtie2 [125]. Similar to the total RNA-seq, HTSeq required the—nonunique all option to accurately quantitate the multicopy snoRNA and miRNA genes,

and DESeq2 was used for differential expression analysis to compare genotypes. A full set of bash scripts for analysis of both total and small RNA-seq data sets including custom annotations are provided at the github repository (https://github.com/KoppesEA/INS-1_PWS_RNA-Seq).

## Supporting information

**S1 Fig. CRISPR/Cas9 genome editing of the PWS-region generates maternal deletion INS-1 lines 5–5 and 5–9. (A)** Map of proximal sgRNA1 and distal sgRNA3 targeting sites. DNA sequences of the sgRNA targeting sites (blue arrows) are shown, with PAM motifs in grey highlight. Other symbols: blue boxes, proximal and distal PWS paternally-expressed genes (also see **Fig 1A**); red box, flanking maternally-expressed gene; black arrows, PCR primers (F, forward; R, reverse); vertical arrows, canonical double-strand break (DSB) position catalyzed by CRISPR/Cas9. **(B-F)** Fluorescence *in situ* hybridization (FISH) with rat BAC probes CH230-114P11 spanning the PWS-IC (see **Methods**; **Fig 1A**) and control CH230-2B12 (outside the PWS-domain) labeled with Orange-dUTP or Green-dUTP, respectively. Representative interphase nuclei or partial metaphases are shown. Yellow arrows, deletion of the PWS-domain. **(B)** 1-color FISH for parental INS-1::mCherry (INS-1) cells transfected with plasmid vectors expressing EGFP (left) or CRISPR (sgRNA1 + sgRNA3)/Cas9 (right). Most INS-1 control cells show diploid PWS BAC (orange) signals, with < 1% of cells having a single PWS-signal due to either a technical artifact (hybridization to a single allele) or to loss of a chromosome (while a small percent of cells shows increased signals due to artifact or to aneusomy). In contrast, genome editing greatly increases the percentage of cells with hemizygosity for the PWS-signal and hence a deletion for the PWS-locus. **(C)** Two-color FISH for control line 3–18, with most cells showing diploid signals for both BAC probes. **(D)** FISH for maternal (MAT)-deletion (del) line 5–5, with virtually all cells showing deletion of the PWS-locus. **(E)** FISH for MAT-del line 5–9 showing mosaicism, one cell population intact (and negative for the mCherry transgene) and one (mCherry-positive) deleted for the PWS-locus. **(F)** FISH on flow sorted mCherry-positive MAT-deletion line 5–9, almost all cells having the PWS-deletion. **(G)** INS-1 deletion lines 5–5 and 5–9 express PWS-imprinted genes. **(H)** Schematic of origin for MAT-deletion INS-1 lines 5–5 and 5–9, based on loss of maternal DNA methylation (**Fig 1D**) and presence of paternal-gene expression data in **S1G Fig** Paternally- (P) and maternally-derived (M) chromosomes shown in blue or red, respectively (del, deletion). **(I)** Gels showing PCR fragments spanning sgRNA1 and sgRNA3 for MAT-del lines 5–5 and 5–9. See **S2C–S2H Fig** for Sanger sequencing data on these PCR fragments. (JPG)

**S2 Fig. Sanger sequencing of genome editing events at sgRNA sites in derivation of maternal deletion INS-1 clonal lines. (A-H)** Sanger sequence traces are shown highlighting sgRNA1 (yellow), sgRNA3 (blue), SpCas9 NGG PAMs (grey), and insertion mutations (pink). **(A-B)** Maternal-deletion allele of clonal lines 5–9 **(A)** and 5–5 **(B)** are identical canonical deletions with 3.16 Mb deleted between sgRNA1 and sgRNA3 due to a breakpoint from DNA repair of DSBs occurring at each position 3-nt upstream of the PAM (PAM-3) nuclease sites (see **S1A Fig**). Deletion-PCR breakpoint fragments are from the **Fig 1C** gel. **(C-E)** Sequence of intact sgRNA1 site in parental INS-1 **(C)**, and scarred alleles with a single T/A insertion at the canonical PAM-3 DSB position (pink highlight) in clonal lines 5–9 **(D)** and 5–5 **(E)**. Sequenced PCR fragments spanning sgRNA1 are from the gel in **S1I Fig** **(F-H)** Sequence of intact sgRNA3 site in parental INS-1 **(F)**, and scarred alleles with a G/C insertion or a C/G insertion at the PAM-3 site (pink highlight) in line 5–9 **(G)** or line 5–5 **(H)**, respectively. Sequenced PCR fragments spanning sgRNA3 are from the gel in **S1I Fig** It may be noted that

as the deletions for 5–9 and 5–5 are on the maternal allele, the sgRNA1 and sgRNA3 scarred alleles can be inferred to occur on the paternal allele for each cell line. Further, as each of lines 5–5 and 5–9 have different sgRNA3 scarred alleles (despite sharing deletion breakpoints and sgRNA1 scarred allele mutations) then these two cell lines clearly arose as independent genome editing events.
(JPG)

**S3 Fig. Sanger sequencing of sgRNA1 and sgRNA3 off-target sites in maternal deletion INS-1 clonal lines.** Sanger sequencing chromatographs of direct sequenced off-target genomic PCRs are shown for the top three ranked predicted off-target sites for sgRNA1 highlighted by the sgRNA seed (yellow) and for sgRNA3 (blue), with SpCas9 NGG PAMs (grey), and deviation in off-target from sgRNA sequence (brown). **(A-C)** Sequence of intact sgRNA1 off-target site at chromosome 2 position 115,451,401–115,451,423 (+; intergenic *Gnb4-Actl6a*) with 1 mismatch in parental INS-1 **(A)** and maternal (MAT)-deletion lines 5–9 **(B)**, and 5–5 **(C)**. **(D-F)** Sequence of intact sgRNA1 off-target site at chromosome 6 position 102,394,187–102,394,209 (-; intron *Rgs6*) with 2 mismatches in parental INS-1 **(D)** and MAT-deletion lines 5–9 **(E)**, and 5–5 **(F)**. **(G-I)** Sequence of intact sgRNA1 off-target site at chromosome 1 position 185,582,894–185,582,916 (+; intergenic *Htra1-Dmbt1*) with 3 mismatches in parental INS-1 **(G)** and MAT-deletion lines 5–9 **(H)**, and 5–5 **(I)**. **(J-L)** Sequence of intact sgRNA3 off-target site at chromosome 19 position 19,693,031–19,693,053 (+; intergenic *Cbln1-N4bp1*) with 3 mismatches in parental INS-1 **(J)** and MAT-deletion lines 5–9 **(K)**, and 5–5 **(L)**. **(M-O)** Sequence of intact sgRNA3 off-target site at chromosome 6 position 88,102,670–88,102,692 (+; intron *Sos2*) with 2 mismatches in parental INS-1 **(M)** and MAT-deletion lines 5–9 **(N)**, and 5–5 **(O)**. **(P-R)** Sequence of intact sgRNA3 off-target site at chromosome 9 position 105,005,251–105,005,273 (+; intron *Tmem232*) with 3 mismatches in parental INS-1 **(P)** and MAT-deletion lines 5–9 **(Q)**, and 5–5 **(R)**. In all off-target sites analyzed for both sgRNA1 and sgRNA3 there was no evidence of CRISPR-Cas9 induced dsDNA break repair resulting in small insertion-deletion events.
(JPG)

**S4 Fig. Further CRISPR/Cas9 genome editing of the PWS-region generates paternal deletion INS-1 lines 3, 19–1, 19–4, and 25. (A)** Map of proximal sgRNA70-3 and distal sgRNA79-1 targeting sites on the paternal (PAT)-allele. Both sgRNA sites are absent on the maternal allele of lines 5–5 and 5–9 (**S1 Fig**), providing the PAT-allele specificity. Further proximal, the alt-sgRNA70-3 site differs at two 5′ nucleotides of the sgRNA and is part of a segmental-duplication (to be described elsewhere; manuscript in preparation). All symbols are as for **S1A Fig** **(B)** Deletion-PCR assays for a CRISPR/Cas9 screen of four pairs of sgRNAs flanking the 3.143 Mb PWS-domain, using control INS-1 cells. EGFP represents control transfections. Of the four pairs of sgRNAs, the highest efficiency is obtained for sgRNA70-3 + sgRNA79-1. **(C)** Deletion-PCR assay following transfection of MAT-del lines 5–5 and 5–9 using the CRISPR (sgRNA70-3 + sgRNA79-1)/Cas9-expressing vector shows specific targeting of the paternal allele of the PWS-domain. The positive (+) control is from **S4B Fig** **(D-F, H-L)** FISH studies using rat FISH probes, as for **S1B–S1F Fig** **(D)** FISH for parental INS-1 cells transfected with plasmid vectors expressing EGFP (not shown) or CRISPR (sgRNA70-3 + sgRNA79-1)/Cas9. Control cells for both probes (not shown) and the control probe (green) show diploid signals in virtually all cells, whereas for the PWS-probe (orange) in genome edited cells 5% of cells display an interphase with a PWS-region deletion (yellow arrow). **(E-N)** Studies on clonal lines derived from transfection of the MAT-del 5–9 cell line with an EGFP control vector **(E-F, N)** or with the CRISPR (sgRNA70-3 + sgRNA79-1)/Cas9-expressing vector **(G-M)**. **(E)** Interphase and partial metaphase FISH for control line 2 with a MAT-deletion in virtually all cells. **(F)**

FISH for control line 16 with a MAT-deletion in most cells. **(G)** Deletion-PCR for PAT-del line 3 using alt-sgRNA70-3 specific F and sgRNA79-1 R primers. As compared to the faint deletion breakpoint band using a sgRNA70-3 F PCR primer (**Fig 1E**), this alternate (alt) assay provides a high degree of specificity, with fainter deletion-PCR bands for lines 19–1, 19–4, and 25 due to mismatches in the F primer flanking the sgRNA70-3 site. See **Fig S5J** for Sanger sequencing data on the line 3 PCR fragment. **(H)** FISH for PAT-del (PWS) line 3 showing homozygous loss of the PWS-region in most cells. **(I)** FISH for PAT-del (PWS) line 25 shows homozygous loss of the PWS-region but mosaicism for cells tetrasomic or disomic for chromosome 1 (PWS-locus). **(J)** FISH for mixed line 19 shows mosaicism with most cells having homozygous loss of the PWS-region but a small % of cells with a MAT-deletion (from the parental 5–9 line) having an intact paternal PWS-region (orange arrow). **(K-L)** Single cells from dilution of mixed line 19 and 96-well plating underwent a further clonal isolation, followed by PCR screening of genomic DNA for loss of PWS-loci (not shown) and FISH, with isolation of 5 sub-lines. **(K)** Metaphase from FISH on PAT-del (PWS) line 19–1 showing homozygous loss of the PWS-region in most cells. **(L)** FISH for PAT-del (PWS) line 19–4 with homozygous loss of the PWS-region in virtually all cells. **(M)** Schematic of origin for PAT-deletion INS-1 lines 3, 19, 19–1 through 19–5, and 25. **(N)** Gels showing PCR fragments spanning sgRNA70-3 and sgRNA79-1 for parental INS-1 and control lines 2 and 16. See **S5A–S5F Fig** for Sanger sequencing data on these PCR fragments. Note that there are no scarred allele sites for sgRNA70-3 and sgRNA79-1 in the paternal-deletion lines 19–1, 19–4 and 25 as the maternal allele is deleted for these loci (see **Figs 1A**, **S1A and S5A**).
(JPG)

**S5 Fig. Sanger sequencing of genome editing events at sgRNA sites in derivation of paternal deletion INS-1 clonal lines. (A-J)** Sanger sequence traces are shown highlighting sgRNA70-3 (green), sgRNA79-1 (pink), and SpCas9 NGG PAMs (grey). **(A-C)** Paternal (PAT)-del allele of clonal line 19–1 **(A)**, 19–4 **(B)**, and 25 **(C)** are identical canonical deletions with 3.143 Mb deleted between the sgRNA70-3 and sgRNA79-1 PAM-3 nuclease sites. Deletion-PCR breakpoint fragments are from gels such as shown in **Fig 1E**. **(D)** PAT-del line 3 generated an unexpected proximal deletion breakpoint at the canonical PAM-3 position of an alternate (alt) sgRNA70-3 site upstream of *Frat3*, that maps ~ 125 kb upstream of the sgRNA70-3 position (see **Figs 1A** and **S4A**). The alt-sgRNA70-3 site differs only at 2 of the most 5′ nucleotides of the sgRNA (purple highlight) and is part of an uncharacterized segmental-duplication. The line 3 genome-editing event with a 3.268 Mb deletion has at the breakpoint a 233- or 234-bp insertion of sequence with 97% identity to *E. coli* (purple sequence, with *gsiD* amber stop codon indicated in red), followed by a distal deletion breakpoint with a single G/C nucleotide insertion or polymorphism in *E. coli* sequence (orange highlight) that occurred at a DSB at the PAM-2 position of the distal sgRNA79-1 site (pink highlight). The deletion-PCR breakpoint fragment for the PAT-del line 3 is from the gel in **S4G Fig** **(E-G)** Sequence of intact sgRNA70-3 site in parental INS-1 **(E)** and maternal (MAT)-deletion (del) control lines 2 **(F)** and 16 **(G)**. Sequenced PCR fragments spanning sgRNA70-3 are from the gel in **S4N Fig** **(H-J)** Sequence of intact sgRNA79-1 site in parental INS-1 **(H)** and MAT-del control lines 2 **(I)** and 16 **(J)**. Sequenced PCR fragments spanning sgRNA79-1 are from the gel in **S4N Fig** **(K-L)** Sanger sequence traces of the alt-sgRNA70-3 site by direct Sanger sequencing of a genomic PCR product showing an intact site in parental INS-1 **(K)**, and a hemizygous scarred allele with a 28-nt deletion (leaving 6 nt intact at the 5'-end of the alt-sgRNA70-3 sgRNA site) in clonal PAT-del line 19–1 **(L)**. Sequence from the biphasic chromatograph portion in **(L)** was determined from PCR cloning and Sanger sequencing of the intact (upper) and scarred (lower) alleles. Control lines 5–9, 2, 16, and PAT-del lines 3 and 25 have the same

intact alt-sgRNA70-3 sequence as INS-1 indicating homozygosity for the control lines and PAT-del line 25 while for PAT-del line 3 this indicates an intact MAT-allele at alt-sgRNA70-3 since that site is involved in the PWS-deletion breakpoint (see **S5D Fig**). As expected, PAT-del line 19–4 has equivalent results to line 19–1, with heterozygosity for an identical scarred allele. (JPG)

**S6 Fig. Copy number of PWS-loci based on droplet-digital PCR (ddPCR) using EvaGreen for PWS *vs.* control INS-1 lines. (A)** Schematic of the PWS-imprinted domain with paternally expressed genes in blue and the maternally expressed *Ube3a* in purple. The positions of the sgRNAs that mark the PWS-deletion breakpoints are indicated by pink boxes, the PWS-IC by a green box, and the four loci examined by ddPCR by yellow boxes. **(B)** Genomic ddPCR 1d amplitude plot for *Ube3a*, localized outside of the distal PWS-deletion breakpoint and hence intact in all INS-1 cell lines. **(C-E)** Genomic ddPCR 1d amplitude plots for **(C)** *Mirh1*, **(D)** *Snurf-Snrpn*, and **(E)** *Snord107*, each localized within the PWS-deletion region. **(F)** Schematic showing generation in an initial CRISPR/Cas9 genome editing screen of clonal control (Con) cell line 5–9 with an ~ 3 Mb deletion of the PWS-domain on the maternal allele, and additional clonal cell lines generated from a second CRISPR/Cas9 genome editing screen. The latter include two further control lines, 2 and 16, each with a PWS-domain deletion on the maternal allele, and three independent homozygous deletion sublines 3, 19 and 25. **(G)** Graph of genomic ddPCR for the PWS-region demonstrating that sublines 3 and 25 are pure populations of cells with a homozygous PWS-region deletion. However, whereas most cells in subline 19 have a homozygous PWS-deletion, there is a small percentage of cells having an intact PWS-region (i.e., derived from the 5–9 parental line) as a mixed population. FISH also confirmed 3.7% of cells with an intact PWS-region in subline 19 (see **S4J Fig**). Note that subline 19 then had a further screen by plating single cells in a 96-well plate and selecting clonal lines with a homozygous PWS-deletion (i.e., 19–1, 19–2, 19–3, 19–4, 19–5). (JPG)

**S7 Fig. Sanger sequencing of sgRNA70-3 off-target sites in maternal deletion and paternal deletion INS-1 clonal lines. (A-N)** Sanger sequence traces are shown for the top 2 ranked predicted off-target sites for sgRNA70-3 highlighted by the sgRNA seed (green), SpCas9 NGG PAMs (grey), and deviation in off-target from sequence from sgRNA70-3 (brown). **(A-G)** Sequence of intact sgRNA70-3 off-target site at chromosome 1 position 178,678,292–178,678,314 (-; intron *Hs3st4*) with 3 mismatches in parental INS-1 **(A)** and maternal (MAT)-deletion lines 5–9 **(B)**, line 2 **(C)**, line 16 **(D)**, and paternal (PAT)-deletion lines 3 **(E)**, 19–1 **(F)** and 19–4 **(G)**. **(H-N)** Sequence of intact sgRNA70-3 off-target site at chromosome 8 position 72,506,613–72,506,635 (+; intron *Tcf12*) with 2 mismatches in parental INS-1 **(H)** and MAT-deletion lines 5–9 **(I)**, line 2 **(J)**, line 16 **(K)**, and PAT-deletion lines 3 **(L)**, 19–1 **(M)** and 19–4 **(N)**. (JPG)

**S8 Fig. Sanger sequencing of sgRNA79-1 off-target sites in maternal deletion and paternal deletion INS-1 clonal lines. (A-U)** Sanger sequence traces are shown for the top 3 ranked predicted off-target sites for sgRNA79-1 highlighted by the sgRNA seed (pink), SpCas9 NGG PAMs (grey), and deviation in off-target from sequence from sgRNA79-1 (brown). **(A-G)** Sequence of intact sgRNA79-1 off-target site at chromosome 9 position 58,847,915–58,847,937 (+; intergenic *Satb2*-RGD1306941) with 2 mismatches in parental INS-1 **(A)** and maternal (MAT)-deletion lines 5–9 **(B)**, line 2 **(C)**, line 16 **(D)**, and paternal (PAT)-deletion lines 3 **(E)**, 19–1 **(F)** and 19–4 **(G)**. **(H-N)** Sequence of intact sgRNA79-1 off-target site at chromosome 3 position 128,425,398–128,425,420 (+; intron *Macrod2*) with 4 mismatches in parental INS-1 **(H)** and MAT-deletion lines 5–9 **(I)**, line 2 **(J)**, line 16 **(K)**, and PAT-deletion lines 3 **(L)**, 19–1

(M) and 19–4 (N). (O-U) Sequence of intact sgRNA79-1 off-target site at chromosome 2 position 47,166,018–47,166,040 (+; intergenic *Itga1-Isl1*) with 4 mismatches in parental INS-1 (O) and MAT-deletion lines 5–9 (P), line 2 (Q), line 16 (R), and PAT-deletion lines 3 (S), 19–1 (T) and 19–4 (U).
(JPG)

**S9 Fig. Copy number of PWS-loci based on droplet-digital PCR (ddPCR) using TaqMan probes for PWS *vs*. control INS-1 lines.** (A) Schematic of the PWS-imprinted domain with paternally expressed genes in blue and the maternally expressed *Ube3a* in purple. The positions of the sgRNAs that mark the PWS-deletion breakpoints are indicated by pink boxes, the PWS-IC by a green box, and the probes used for two loci that were examined by TaqMan ddPCR by orange boxes. (B) ddPCR 2d plots for TaqMan probe copy number assay with absorbance amplitude for channel 1 (*Snord107*, FAM) on the y-axis and channel 2 (*Ube3a*, HEX) on the x-axis. Blue dots denote *Snord107* positive droplets, green dots represent *Ube3a* positive droplets and orange indicate double positive droplets. Note the absence of channel 1 *Snord107* positive (blue or orange droplets) in the PWS lines, and 50% reduction of channel 1 positive droplets in maternal-deletion control (Con) lines (also see **Fig 1F** for graphical data). NTC, no template control.
(JPG)

**S10 Fig. Gene expression for PWS-imprinted genes, *Ube3a* and *Ube3a-ATS* loci.** (A) RT-PCR analyses of 7 PWS-imprinted genes (*U1A*, *Snurf*, *Snrpn*, *Snord107*, *Snord64*, *Snord116*, *Snord115*; see map in **Fig 1A**), rat *Ins2*, *mCherry* transgene, and *Gapdh* control gene in the expanded INS-1 panel of 9 cell lines. U1 represents an alternate upstream (U) promoter-first exon that splices into *Snurf-Snrpn* exon 2 and has multiple duplicate U1 copies in rodents. Abbreviations: ex, exon; RT-, PCR control using RNA not treated with reverse-transcriptase. The absence of bands in the RT- assay for the multicopy, tandemly repeated *Snord116* locus rules out genomic DNA contamination in the RNA. Note that the RT-PCR gel results for 4 genes (*Gapdh*, *Ins2*, *Snrpn* ex 8–10, *Snord116*) in the 6-cell line panel used in this entire study are reproduced from **Fig 1G** for direct comparison to the remaining genes in the PWS domain that are presented here, plus this dataset has 3 additional clonal PWS INS-1 cell lines (19–2, 19–3, 19–5). Similarly, while the *Snurf* ex 1–3 results are shown here for the expanded panel of nine INS-1 cell lines, these same results for the 6-cell line panel used in this entire study are reproduced in **S11B Fig** for comparison to other loci examined in that dataset. (B) RT-PCR analyses of 2 PWS-imprinted genes (*Mkrn3*, *Ipw*) as well as *Ube3a* and *Ube3a-ATS* loci (see maps in **Figs 1A** and **S10C**), and control genes (*Gapdh*, *Sim1*) in the 6-cell line panel used in this entire study. The upper 6 rows of gels are from standard RT-PCR assays that use random hexamer primers for the RT primer and typical gene-specific RT-PCR primers, while the lower 3 rows of gels are from RT-PCR assays that use a strand-specific primer for RT followed by RT-PCR with either a *Ube3a* (ex12 F + ex13 R) or *Ube3a-ATS* (ex13 R + int12 F) primer set. Combined, the data show that transcripts from the *Ube3a-ATS*-region that are detected when using random primers for RT derive from *Ube3a* primary (1°) transcripts since there is no expression from the *Ube3a-ATS* strand. (C) Map of the *Ube3a* and *Ube3a-ATS* locus. Long arrows represent transcriptional orientation and extent, with *Ube3a* spliced (exon 12–13), and *Ube3a-ATS* transcribed as the 3'-end of long lncRNAs from the *U1A-Snurf-Snrpn*-snoRNA locus. While *Ube3a* is highly expressed in all INS-1 cell lines, as seen in **S10B Fig** there is no detectable expression of *Ube3a-ATS* in control or PWS cell lines. Abbreviations: del brkpt, CRISPR/Cas9 deletion breakpoint in PWS INS-1 cell lines; ex, exon; int, intron; short arrows, PCR primers; parentheses [(,)] indicate the lack of *Ube3a-ATS* transcripts.
(JPG)

**S11 Fig. Expression of a ψ*Snurf*-ψ*Snrpn* locus within the rat *Mon2* gene. (A)** RNA-seq analysis using a custom rat genome build identifies a ψ*Snurf-Snrpn* locus expressed in PWS and control INS-1 lines. In contrast, the imprinted *Snurf* and *Snrpn* loci are only expressed in the control INS-1 lines. Expression for the pseudogene host gene, *Mon2*, is also shown. **(B)** RT-PCR with gel analysis for 3 amplicons from *Snurf* as well as ψ*Snurf*-ψ*Snrpn*, *Mon2* and *Atp10a* in the INS-1 panel. The first 2 rows show primer sets designed to amplify *Snurf* exons (ex) 1–2 and 1–3, while row 3 shows RT-PCR with a primer set designed to amplify both *Snurf* and ψ*Snurf* sequences; the latter RT-PCR products were then digested with the *Pml*I restriction endonuclease which distinguishes between *Snurf* and ψ*Snurf* products. Note that *Atp10a* is significantly reduced in PWS *vs*. control lines (also see **S1** and **S2 Tables**). **(C)** Map location of the expressed ψ*Snurf*-ψ*Snrpn* locus (green box) within the brown rat (*Rattus norvegicus*) *Mon2* gene (brown boxes are 5' exons 1–3). In contrast, the black rat (*Rattus rattus*) has no pseudogene insertion and a single copy of the Target Site Duplication (TSD, purple arrows) present in the brown rat genome at the 5'-end of the inserted pseudogene. Black arrows represent PCR primers, while the underlined TAA nucleotides represents a tandem duplication in the 3'-TSD copy. **(D)** DNA sequence of ψ*Snurf* transcripts in PWS INS-1 lines compared to the endogenous *Snurf* gene. Green shading of red nucleotides indicates missense mutations, and blue shading with a hyphen represents a nucleotide deletion. Orange font represents the *Pml*I cleavage site specifically in the *Snurf* cDNA sequence. Snurf start and stop codons are indicated. **(E)** DNA sequence identifies a frameshift from a single nucleotide deletion and premature stop codon in the 5' ψ*Snrpn* portion of the pseudogene. Symbols as for **(D)**. **(F)** DNA sequence identifies a 21-nucleotide deletion in the 3' ψ*Snrpn* portion of the pseudogene. Symbols as for **(D)**. **(G)** Potential ψSnurf amino acid sequence encoded by the ψ*Snurf* gene. Green or blue highlight represents potential amino acid changes or in-frame deletion, respectively, and red asterisks the stop codons. **(H)** Potential truncated ψSmN amino acid sequence encoded by the ψ*Snrpn* gene. Symbols as for **(G)**, except blue highlight, amino acids resulting from a frame-shift mutation. Assuming no evolutionary selection for function, we can date the age of rat ψ*Snurf*-ψ*Snrpn* based on a total of 26 mutations/substitutions in 1,288-nt of homologous sequence to rat *Snurf-Snrpn*. The 2.02% divergence (97.98% similarity) corresponds using a mutation rate formula for silent site substitutions and intronic sequences of $5 \times 10^{-9}$ to $7 \times 10^{-9}$ mutations/site/year [126] to a pseudogene origin ~ 2.88–4.0 million years ago (mya), which correlates well with prior molecular dates for divergence of black and brown rats about 2.0–2.9 mya [127,128]. (JPG)

**S12 Fig. Transcriptome analyses in PWS-deletion *vs*. control INS-1 lines for differentially expressed genes (DEGs) and non-DEGs. (A)** Heatmap clustergram of 228 DEGs demonstrates tight clustering of PWS *vs*. control groups (*Padj* < 0.05). Scale: green (enriched) to red (depleted). RNA-seq was performed for 3 PWS (3, 19–1, 19–4) *vs*. 3 control (5–9, 2, 16) INS-1 cell lines. **(B-E)** Quantitative gene expression analyses for control (Con, black; 5–9, 2, 16) *vs*. PWS (red; 3, 19–1, 19–4) determined by RT-ddPCR normalized to *Gpi* levels and to the average expression in control INS-1 lines, for **(B-E)** panels of additional candidate DEGs. The latter include validation of **(B)** 10 down-regulated genes (*Derl3*, *Mylip*, *Atp10a*, *Jph3*, *Ndrg4*, *Tap1*, *Tap2*, *Syt1*, *Robo2*, *Id4*), as well as **(C)** for a set of 4 upregulated DEGs involved in the secretory pathway (*Chgb*, *Cacna1a*, *Tmem176a*, *Tmem176b*), with apparent upregulation for an exogenous human insulin (*Hs*, *Homo sapiens*)-neomycin resistance (*Neo*R) transgene in the INS-1 lines, although the latter is artifactual due to transgene silencing in line 16 (see **S16A, S16I–S16K** and **S16M Fig**). **(D,E)** However, the RT-ddPCR analyses failed to validate **(D)** several candidate upregulated DEGs from the RNA-seq data (**Fig 4A**), or **(E)** candidates for

downregulated DEGs based on data from PWS iPSC-derived neuronal and *Snord116*-deficient mouse models [60,129]; likewise, another mouse study [130] did not confirm *Pcsk1*. Statistical comparison by Welch's t-test: *, *P* < 0.05; **, *P* < 0.005; ***, *P* < 0.0005.
(JPG)

**S13 Fig. Genome-wide snoRNA and miRNA transcriptome changes in PWS INS-1 lines from small RNA-seq. (A)** Heatmap clustergram of 58 differentially expressed miRNAs and snoRNAs demonstrates tight clustering of PWS *vs*. control groups (*P*adj < 0.1). Scale: green (enriched) to red (depleted). Small RNA-seq was performed for 3 PWS (3, 19–1, 19–4) *vs*. 3 control (5–9, 2, 16) INS-1 cell lines. **(B)** Normalized expression counts for the top 14 significant differentially expressed miRNAs and snoRNAs, of which only 3 miRNAs identified are not encoded within the PWS-imprinted domain. Box charts of control (blue) and PWS (red) genotypes are shown with underlying data points for each sample. For the multicopy *Snord115*, *Snord116*, and *Mir344* genes the data for the paralogs were binned.
(JPG)

**S14 Fig. Gene ontology enrichment of DEGs. (A)** EnrichR gene ontology (GO Biological Process 2018 version) terms enriched in down-regulated DEGs highlighting ER stress mediators including IRE1, ATF6, and ER-associated protein degradation (ERAD) pathways. **(B)** DAVID analysis reveals a functional annotation cluster (Enrichment Score: 4.376) and **(C)** considerable overlap of ER functions and unfolded protein response components among the functional annotation groups (as represented by the green boxes). Gene ontology analysis performed on a gene set of 81 downregulated DEGs filtered by adjusted *P* value < 0.1 and Fold Change < 0.75 (log2FC < -0.415) with all PWS genes removed.
(JPG)

**S15 Fig. Transcription factor binding site enrichment in the promoters of DEGs. (A-B)** Enrichr transcription factor (TF) analysis of 94 down-regulated non-PWS DEGs (*P*adj < 0.10 and an absolute fold-change (FC) >1.25) shown as clustergrams of **(A)** top 20 ENCODE and ChEA Consensus TFs from ChIP-X or **(B)** top 30 TRANSFAC and JASPAR PWMs enriched in the gene set. **(C-F)** Enrichr TF analysis for 20 RT-ddPCR validated down-regulated genes (from **Figs 4D, 4E** and **S12B**: *Hspa5, Hsp90b1, Pdia4, Pdia6, Ppib, Creld2, Sdf2l1, Dnajb11, Dnajc3, Hyou1, Npy, Iapp, Derl3, Mylip, Atp10a, Jph3, Ndrg4, Tap1, Tap2, Syt1*. Further analysis of validated DEGs enriched for TF binding sites from **(C)** ENCODE and ChEA Consensus TFs, **(D)** TRANSFAC and JASPAR motifs, **(E)** TF Perturbations Followed by Expression (DE, differential expression; KD, knockdown; KO, knockout; OE, over-expression) with GEO accession listed, and **(F)** TF-LOF Expression from GEO with PMID reference appended (LOF, loss of function). Clustergrams generated based on enrichment *P*-value significance which shows as the red highlight of TF names. Notably in the downregulated genes in PWS INS-1 cells a prominent cluster of genes including many ER chaperones were enriched for ATF6-cofactor NFYA and NFYB binding sites and other potential regulators including CPEB1, RFX5, IRF3, CREB1 and SREBF1 (see **S15A–S15D Fig**). Similarities to other model systems revealed analogous gene-expression changes including for *Xbp1* perturbations in an adipose cell line [131] and the *Pparb/d* KO mouse pancreas [93] (see **S15E–S15F Fig**).
(JPG)

**S16 Fig. Specificity of RT-PCR and RT-ddPCR assays for rat, mouse and human insulin genes in the INS-1 cell line panel. (A)** RT-PCR with gel analysis for insulin genes. Abbreviations are: (1), (2), represent amplicons 1 and 2 for a given gene; mCherry, INS-1(832/13):: mCherry cell line; 832/13, INS-1(832/13) cell line; E, INS1-E cell line; *Hs*, *Homo sapiens*; *Mm*, *Mus musculus*; *Rn*, *Rattus norvegicus*; *, non-specific band in the indicated RT-PCR assay

(note that this amplified product is not present using the same PCR primer pair in the RT-ddPCR assay shown in **S16J Fig**, likely due to different chemistry or annealing temperature in the two assays). Note that control line 16 has greatly reduced expression of the *Hs INS* (amplicons 1 and 2) and *NeoR* segments of the *INS-NeoR* transgene, as also seen in the RT-ddPCR data [see **S16–S16K Fig**] and RNA-seq data [see **S16M Fig**], likely reflecting epigenetic inactivation of the *INS-NeoR* transgene in a majority of cells for line 16 (as DNA analysis indicated the transgene remained present). **(B-L)** RT-ddPCR assays for the listed amplicons, with ddPCR performed using EvaGreen. All abbreviations are as for **S16A**. **(B)** RT-ddPCR assay for *Rn Ins1* amplicon 1. **(C)** RT-ddPCR assay for *Rn Ins1* amplicon 2. **(D)** RT-ddPCR assay for *Rn Ins2* amplicon 1. **(E)** RT-ddPCR assay for *Rn Ins2* amplicon 2. **(F)** RT-ddPCR assay for *Mm Ins2*. **(G)** RT-ddPCR assay for *mCherry* amplicon 1. **(H)** RT-ddPCR assay for *mCherry* amplicon 2. **(I)** RT-ddPCR assay for *Hs INS* amplicon 1. **(J)** RT-ddPCR assay for *Hs INS* amplicon 2. **(K)** RT-ddPCR assay for *NeoR*. **(L)** RT-ddPCR assay for control gene *Gpi*. **(M)** RNA-seq analysis of insulin genes expressed in the INS-1 lines. In addition to the endogenous rat (r) *Ins1* and *Ins2* genes, a custom rat genome build identified mouse (m) *Ins2-mCherry* and *Hs INS-NeoR* transgene mRNA levels. Red numbers indicate that control line 16 is an outlier with drastically reduced expression of the *Hs INS-NeoR* transgene (see **S16A Fig** legend). (JPG)

**S17 Fig. Electron microscopy establishes normal subcellular organelles in PWS and control INS-1 lines. (A)** Control line 5–9, **(B)** Control line 2, **(C)** Control line 16, **(D)** PWS line 3, **(E)** PWS line 19–1, **(F)** PWS line 19–4. The scale bar in the bottom right corner of each image is 800 nm, while abbreviations for features are: N, nucleus; M, mitochondria; rER, rough endoplasmic reticulum (for M and rER, only one label per image is shown). Key subcellular organelles of normal appearance include mitochondria and rough ER. These INS-1 832/13-derived cell lines, while producing and secreting significant insulin [34] (**Figs 2–5**) and other secretory peptides such as IAPP (**Fig 3C and 3D**) do not demonstrate clearly visible insulin secretory vesicles, likely because of the relative rarity of these compared to a number in the tens of vesicles detected in the INS-1E cell line [132] that produces much more abundant insulin (see **Fig 5A**). (JPG)

**S18 Fig. PWS INS-1 β-cell lines are more sensitive to thapsigargin-induced ER stress with earlier and more robust activation of *Xbp1* "splicing". (A)** Reverse transcription-PCR gel electrophoresis analysis of *Xbp1* mRNA processing for exclusion of 26-nt of exon 4 from time 0 to 5 hours of thapsigargin treatment. Abbreviations: S, spliced; T, total; U, unspliced. **(B)** Time-course of *Xbp1* mRNA activation as the ratio of spliced/total mRNA detected by RT-PCR in **(A)**. Initially, at 1 hr of thapsigargin treatment *Xbp1* mRNA levels fall due to mRNA turnover for both control and PWS cell lines. *, $P < 0.05$ as calculated by ANOVA. (JPG)

**S19 Fig. PWS INS-1 β-cell lines are more sensitive to tunicamycin-induced ER stress with earlier and more robust activation of ATF6-N. (A)** Time-course western blots of ATF6 in PWS and control cell lines treated with tunicamycin for 0, 1, 2, 3, 4, 5, 6 or 8 hours. The ratio of processed nuclear (N) isoform over full-length unglycosylated (FL-UG) is written below each lane. FL-G, full-length glycosylated; *, non-specific band detected by the anti-ATF6 antibody. **(B)** Relative ratio of each ATF6 isoform as a fraction of the total ATF6 during the time-course of tunicamycin treatment. Line graphs represent mean of three measurements at each time point, except n = 6 using technical replicates for 4 hr and n = 6 using two biological replicates for 5 hr timepoints, for full-length glycosylated (FL-G) over total (Red); FL-UG over total (Blue) and N over total (Green) with control (Con) as solid lines and PWS cell lines as dotted

lines. Data indicates that there is a clear trend to more robust processing of FL-UG to N at early tunicamycin timepoints in PWS cell lines.
(JPG)

**S1 Table. Differentially expressed genes (DEGs) in PWS *vs.* control INS-1 lines from RNA-seq with HTSeq feature counts.** Data presented from RNA-seq expression bioinformatics pipeline using a custom rat annotation build with STAR aligner, gene level counts with HTSeq (with options—mode union and—nonunique all) and differential expression with DESeq2. Top DEGs with a cutoff of *P*adj < 0.1 are shown. Full gene expression table available at GEO repository GSE190334.
(XLSX)

**S2 Table. Differentially expressed genes (DEGs) in PWS vs. control INS-1 lines from RNA-seq with RSEM feature counts.** Data presented from RNA-seq expression bioinformatics pipeline using a custom rat annotation build with STAR aligner, gene level counts with RSEM and differential expression with DESeq2. Top DEGs with a cutoff of *P*adj < 0.1 are shown. Full gene expression table available at GEO repository GSE190334.
(XLSX)

**S3 Table. Significant differentially expressed small RNAs (DE sRNAs) in PWS *vs.* control INS-1 lines from small RNA-seq.** Data presented from small RNA-seq expression bioinformatics pipeline, deduplicated with UMI Tools, mapped using a custom rat annotation build with Bowtie2 aligner, with gene level counts quantified by HTSeq (with option—nonunique all) and differential expression with DESeq2. Top DE sRNAs with a cutoff of *P*adj < 0.1 are shown. PWS sRNAs gene names are color coded *Snord116* (orange), *Snord115* (blue), *Mir344* (red), *Snord64* (green) and *Snord107* (purple). Full gene expression table available at GEO repository GSE190336.
(XLSX)

**S4 Table. Highly expressed miRNAs in INS-1 lines from small RNA-seq.**
(XLSX)

**S5 Table. Highly expressed snoRNAs in INS-1 lines from small RNA-seq.**
(XLSX)

**S6 Table. sgRNA oligonucleotides.** *Bbs*I site oligonucleotide cloning adapters for sgRNAs. Abbreviations: b/w, between; F, forward.
(XLSX)

**S7 Table. Genomic PCR, copy number ddPCR, and DNA methylation PCR primers.** Abbreviations: F, forward; R, reverse; un, unmethylated; me, methylated; IC, imprinting center; chr, chromosome.
(XLSX)

**S8 Table. RT-PCR and RT-ddPCR primers.** Abbreviations: F, forward; R, reverse; RT-PCR, reverse transcription-PCR; RT-ddPCR, reverse transcription-droplet digital PCR.
(XLSX)

**S9 Table. Antibodies used in this study.**
(XLSX)

**S1 Data. All numerical and statistical data for quantitation figures.**
(XLSX)

## Acknowledgments

We thank Dr. Michael R. Rickels and Heather W. Collins at the University of Pennsylvania Diabetes Research Center RIA/Biomarkers Core (NIH P30-DK019525), Mary F. Sanchirico for the protein database, Dr. Régis A. Costa for off-target PCR primer design, and Drs. Rebecca A. Simmons and Dwi Kemaladewi for review of the manuscript. This project used the University of Pittsburgh Health Sciences Sequencing Core at UPMC Children's Hospital of Pittsburgh, for RNA sequencing. This manuscript is dedicated to PWS families and especially the Storr family for their support of our PWS research program.

## Author Contributions

**Conceptualization:** Erik A. Koppes, James J. Moresco, Peter Drain, Robert D. Nicholls.

**Data curation:** Erik A. Koppes, Marie A. Johnson, James J. Moresco, Dale W. Lewis, Donna B. Stolz, Peter Drain, Robert D. Nicholls.

**Formal analysis:** Erik A. Koppes, James J. Moresco, Hyun Jung Park, Peter Drain.

**Funding acquisition:** John R. Yates, III, Robert D. Nicholls.

**Investigation:** Erik A. Koppes, Marie A. Johnson, James J. Moresco, Patrizia Luppi, Dale W. Lewis, Donna B. Stolz, Jolene K. Diedrich.

**Methodology:** Erik A. Koppes, James J. Moresco, Simon C. Watkins, Susanne M. Gollin, Peter Drain.

**Project administration:** Robert D. Nicholls.

**Resources:** Ronald C. Wek.

**Visualization:** Erik A. Koppes, Robert D. Nicholls.

**Writing – original draft:** Erik A. Koppes, Robert D. Nicholls.

**Writing – review & editing:** Erik A. Koppes, Marie A. Johnson, James J. Moresco, Patrizia Luppi, Dale W. Lewis, Donna B. Stolz, Jolene K. Diedrich, John R. Yates, III, Ronald C. Wek, Simon C. Watkins, Susanne M. Gollin, Hyun Jung Park, Peter Drain, Robert D. Nicholls.

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
