## [Decision Letter · Decision Letter 0]

13 Oct 2022

Dear Dr Nicholls,

Thank you very much for submitting your Research Article entitled 'Insulin secretion deficits in a Prader-Willi syndrome β-cell model are associated with a concerted downregulation of multiple endoplasmic reticulum chaperones' to PLOS Genetics. Our apologies for the long delay in reaching a decision which was due to difficulty in securing a suitable associate editor and multiple reviewers.

The manuscript has now been fully evaluated at the editorial level and by three independent peer reviewers. As you will see, the reviewers express mixed enthusiasm for the work, recommending rejection, major revision, and minor revision. Although the recommendations are different, there is general agreement about both strengths (independent replication, unbiased and systematic approach, and overall rigor) and weaknesses--the small effect sizes together with the use of a transformed cell line call into question the physiologic relevance of the findings.

The manuscript and the reviews have now been considered by members of the editorial board. Overall, we are positive about moving forward but ask that the concerns raised by reviewers #2 and #3 be addressed in a revised manuscript. While we agree with the concern raised by reviewer #2 point #4 regarding validation in tissue sections or islets from a mouse model, we appreciate that such an endeavor is likely beyond the scope of the current work; however, we ask that you seriously consider and attempt to experimentally address one or more of points #1-3 from reviewer #2. In addition, we ask that all of the minor concerns raised by reviewers #2 and #3 be addressed by changes to the presentation in a revised manuscript. 

We therefore ask you to modify the manuscript according to the review recommendations. Your revisions should address the specific points made by each reviewer. In addition we ask that you provide a detailed list of your responses to the review comments and a description of the changes you have made in the manuscript.

We hope to receive your revised manuscript within the next 60 days. If you anticipate any delay in its return, we would ask you to let us know the expected resubmission date by email to plosgenetics@plos.org.

Yours sincerely,

Gregory S. Barsh

Editor-in-Chief

PLOS Genetics

Scott Williams

Section Editor

PLOS Genetics

Reviewer's Responses to Questions

**Comments to the Authors:**

Reviewer #1: The manuscript by Koppes et al describes experiments in the INS-1 rat beta cell line seeking to understand whether there is an underlying beta cell defect in PWS. As the authors point out, the hormonal defects in PWS are complicated, with potentially interacting defects at the level of hypothalamus, pituitary and end organ hormone producers such as beta cells. In human PWS, the most prominent physiology is hyperphagic obesity and GH/pituitary deficiency. Diabetes sometimes occurs but is thought to be secondary to marked early onset obesity. Although this manuscript contains a large amount of carefully performed work, this reviewer is not sure that the results shed meaningful light on the biology of PWS

1. Rationale for the study is weak: beta cell functional deficiency does not seem to be a prominent feature of PWS

2. Studies only performed in a rat transformed insulinoma cell line are relevant to a human condition. Validation in the mouse model would increase relevance but the perinatal lethal hypoglycemia in the mouse model already differs strongly from the human condition

Reviewer #2: In this study, the authors sought to understand the cell-autonomous mechanisms that may underlie beta cell dysfunction observed in Prader-Willi syndrome (PWS), which includes hypoinsulinemia as a known feature. Using CRISPR, the authors deleted a genomic segment associated with paternal gene imprinting that defines PWS in a modified INS1 beta cell line. The authors conclude that PWS limits expression of ER chaperones, which thus limits the beta cell’s insulin production capacity. This is an interesting concept that emphasizes the field’s increasing understanding of how ER function plays a direct role in the pathogenesis of many forms of diabetes and may ultimately reveal novel mechanisms for modification of ER function. The study benefits from non-biased approaches (e.g. mass spectrometry and RNA sequencing) to broadly evaluate the consequences of the PWS in beta cells. However, there are some limitations that question the broader application of these findings to a clinical setting. These include the sole use of the INS1 cell line, which does not precisely reflect the biology of islet beta cells (in part for the expression of insulin genes from multiple species) and the lack of investigation into whether the findings here would be replicated in a more physiologic setting. The writing overall is organized and easy to understand, though the precise mechanism underlying the observations remains speculative.

Major Issues:

1. The mechanism proposed to explain defects in PWS beta cells seems to be downregulation of ER folding proteins that subsequently impairs proinsulin production/maturation. A final schematic (with known and unknown mechanistic links) would improve clarity. The overarching question, however, seems to remain unanswered: how does reduction in critical ER proteins limit insulin production, in the absence of other evidence of ER dysfunction (e.g., overt activation of the UPR, ER dilation, or insulin misfolding/aggregation)? Can these findings be replicated by directly reducing expression of some of the key ER proteins in question?

2. The presence of multiple insulin genes in this cell line may confound the results, as the changes in insulin gene or protein expression may be different than that in native islet beta cells. More importantly, there is little discussion regarding the presence or absence of higher molecular weight forms of misfolded/aggregated insulin in the PWS cells. In the setting of greatly reduced ER chaperones, one might hypothesize that insulin misfolding and aggregation within the ER would be a likely contributor to impaired insulin maturation and cell dysfunction. Were such forms found in the mass spec or Western blot data? And if not, what would be the explanation?

3. It is quite striking that the PWS cells do not show increased UPR activation at baseline with such a significant reduction in ER chaperones. Is there a difference in expression of UPR sensors, or other reason to help explain a blunting in response? Additionally, it is unclear if the slightly more rapid (but still limited) activation of UPR pathways in stressed PWS cells has clinical relevance, particularly as this was only observed following treatment with potent ER stress activators (thapsigargin and tunicamycin). Does inhibition of UPR signaling improve the cells’ ability to produce mature insulin? Was there an attempt to evaluate the response to more physiologic stressors in these cells, such as palmitate or high glucose?

4. This manuscript would be greatly improved if the main findings were formally evaluated in a more physiologic model, such as tissue sections or isolated islets from TgPWS mice. In this draft, the explanation for differences with previous findings in TgPWS mice is insufficient (Discussion lines 409-419).

Minor Issues:

1. Figure 3 readability would be greatly improved with larger text sizes and the addition of headings for the graphs in Β-D. Also, Figure 3A and the legend should be modified to describe B through D in order.

2. In Figure 5A, there are several unlabeled bands. What insulin forms would these be? Does this insulin antibody have equal affinity to the different species of insulin?

3. The electron microscopy data (Supplemental Figure 17) would be improved with a formal evaluation of insulin granule number and size, though it appears that the current images do not provide adequate representation of granule morphology. Is such data available?

4. Supplemental Figure 18E-F seems to show a larger cohort of cells in these PWS lines with reduced mCherry expression, in disagreement with authors’ conclusion that C-peptide-mCherry expression is unchanged in PWS cells (Figure 5B, E). Does the construct only show mCherry fluorescence after insulin processing (in contrast to the mCherry found in both proinsulin and cleaved C-peptide on Western blot)? This should be clarified, or more representative images chosen.

Reviewer #3: This manuscript investigates the cellular basis of reduced insulin secretion in Prader-Willi syndrome (PWS) by generating several independent INS-1-derived cell lines to model PWS. These PWS model cells have reduced insulin secretion and reduced levels of insulin and several other secreted peptides. A possible mechanism is suggested by transcriptomic and proteomic analyses, demonstrating the reduced expression of a number of endoplasmic reticulum (ER) chaperones that may be important for proper folding of insulin; reduced levels of chaperones may lead to reductions in insulin levels, though it should be emphasized that this is just a model and hasn’t been tested. The experiments are performed well and the data are solid and convincing, though the effect sizes are of rather small magnitude and it is unclear how important the small effects seen here are to the pathophysiology of PWS. The extensive and rigorous molecular characterization of the mutations in the PWS model cell lines is a particular strength, and the fact that several independent PWS and control cell lines are generated increases the confidence in the results. The proteomic and transcriptomic datasets generated in this work are important contributions to the field. We have a number of relatively minor critiques, many related to the writing and presentation of the work.

Specific comments:

1. Though the data appear to be solid, virtually all of the effects are of small magnitude, < 2 fold (e.g. insulin secretion, altered expression of chaperones at both the mRNA and protein levels, ER-stress sensitivity). This is fine, but is not readily apparent from the way the paper is written (e.g. line 400, where insulin secretion is described as “dramatically reduced” or elsewhere where effect sizes can only be gleaned by careful examination of the figures). More transparency and explicit discussion of the effect sizes would be helpful. For instance, it would be helpful to compare the effect sizes seen here to those of PWS mouse models and human patients.

2. The take-home model of the paper is that the effect on insulin in PWS is due to an indirect effect on chaperones. This is a reasonable and interesting model, but given that the magnitude of the downregulation of these chaperones is actually quite small (appears to be less than 2-fold for most or all of them), it seems possible that some other mechanism is at play and this may be a red herring. It is unclear whether such a modest decrease in multiple chaperones would produce the observed effects on insulin content and secretion, though it is an interesting question for future work. However, it would also be nice to see the full lists of proteomic and transcriptomic data presented as supplementary tables for interested readers so possible alternative targets can be more easily explored.

3. It is concluded that PWS cells are unable to compensate for decreases in chaperones because many chaperones are simultaneously downregulated. This argument does not make a lot of sense to us, and would seem to depend on the mechanism of compensation, which is not further described here. For example, if chaperone genes are transcriptionally downregulated in PWS mutants, what precludes an independent compensation mechanism from simply turning transcription back up, unless the PWS genes are important for the compensation process itself? It would help to present more about what is known about this compensation mechanism, and whether it occurs transcriptionally or posttranscriptionally. A small decrease in many chaperones does not inherently seem to preclude a possible compensation mechanism.

4. The paper is rather difficult to read with lots of jargon and a poor narrative flow. The reader has to do a lot of work to figure things out on their own without much help. Several examples of this are provided in the next few comments(#5-8).

5. Some genes or proteins come up quite suddenly without mentioning their functions or significance. For example, in lines 96-101, SNRPN, SNORD116 and SNORD107 have not been introduced yet as PWS genes, which makes the subsequent conclusion confusing that PWS genes function in beta cells.

6. In several places in the proteomics and transcriptomics sections, there are long lists of genes or proteins with very little context to orient the reader. It is hard for the reader to make much of these lists, and some guidance as to why they are considered worth pointing out or short take-home messages in these sections would be useful. For just a couple examples of this see lines 287-288, lines 267-268.

7. The description of engineering PWS INS-1 cells is quite hard to follow. Figure 1B is not very intuitive. These sections demanded a lot of work from the reader, much of which required looking at supplementary figures to understand the main Results sections. As many readers may not look at these figures, it would help to make this section more accessible.

8. The rationale for performing RNA-seq of small RNAs is not provided. This section ends up interrupting the main narrative and feeling tangential.

9. Since PWS cells have altered levels of many proteins, it is unclear whether the total protein content is a good parameter to use for normalization of insulin secretion.

10. It would help to see the unnormalized raw data for the insulin secretion experiments. Figure 2 shows pooled data from several cell lines. It would also be helpful to see the data for each line separately in a supplementary figure.

11. It is not stated how many times proteomic and transcriptomic experiments were replicated. It is stated that each was performed on three control and three PWS cell lines, but it is unclear if each line was tested just once. It seems likely that the data depicted in the figures are pooled from the different lines though this is not stated explicitly. More clarity on these points would be useful. Separate figures for unpooled data of each cell line would also be useful in the Supplement so that variability between lines can be seen.

12. The study emphasizes the deficits in secreted peptides and ER chaperones but doesn’t provide an explanation for proteins that are increased. A number of neuronal active zone proteins are reported to have increased expression at the mRNA level, but for most it is unclear whether this effect extends to the protein level (only CHGB is labeled in Figure 3). The possible relevance of these changes is also unclear. It is pointed out that many of these proteins may play a role in insulin secretion, but it is unclear why potentially increased levels would lead to the decreased secretion observed in PWS cells unless these factors are negative regulators of insulin secretion (though that seems unlikely given their neuronal functions). Thus, the relevance of these results is unclear.

13. It would be helpful to explicitly state in the Results how many of the genes with reported changes in RNA levels were validated and were not validated by RT-PCR experiments (e.g. line 308).

14. It is unclear whether it is standard practice to use an anti-KDEL antibody in Western blots to specifically identify GRP94 and GRP78, given that this antibody would be expected to recognize many proteins. If so, it would be helpful to cite other articles that validate this method or state the same thing.

15. Electron microscopy images in Fig S17 show one picture of each cell line, leading to the conclusion that PWS cells have normal ultrastructure. It is unclear what criteria were used to make this apparently subjective conclusion (no quantitative data are presented). Also, there is no mention of how many cells and sections were examined.

16. Fig S18: confocal cell images are difficult to assess. It would be helpful to zoom in on one cell for better comparison. As with EM data, it is unclear what criteria were used to compare the PWS cells to control and no quantitation is provided, nor is there mention of number of cells examined.

17. It would be helpful to provide the numbers of the specific PCR primers used in Figs 1, S1 & S4.

18. Figs S13B & C are hard to read. They are small and blurry, even when enlarged. The meaning of green boxes in Fig S13C is not explained.

19. The bioinformatic analysis of transcription factor binding sites shown in Fig S14 is not described in the Methods. The legend to this Figure has a number of undefined acronyms.

20. Line 154: it is unclear what is meant by “biparental DNA methylation.” This was confusing because we thought it is only the maternal allele that is methylated. Or is this altered in cell lines? Related, it would be helpful to better describe the nature of the two alleles in the cell line because one does not usually think of the chromosomes in a cell line as being of maternal or paternal origin; are methylation and imprinting retained in the cell line?

21. Line 33-34: because of the word “or”, this sentence is confusing and suggests that the PWS cell line has a deletion of only the paternal allele, when in fact it has deletions of both maternal and paternal alleles.

22. For each sequencing chromatogram, it would be helpful to state whether the PCR product was sequenced directly or cloned first.

23. Line 170: it is unclear why PWS line 3 gives a larger PCR product given that it is a larger deletion. Knowing which PCR primers were used might help.

24. The eIF2a P/T ratios shown in Fig 6B don’t seem to agree with the values plotted in Fig 6E.

25. Fig 1C: What is mixed pop?

26. Fig S9B: What is NTC?

27. Line 251: What is Snord109? It is not shown in Fig 4B or in Fig 1A.

28. Fig S4A legend: typo. It should read “distal sgRNA79-1”

29. Fig S19A typo: the control 2, 1 hr time point seems like it should be 0.03 instead of 0.3

Reviewed (and signed) by Michael Ailion and Chau Vuong

**Have all data underlying the figures and results presented in the manuscript been provided?**

Reviewer #1: Yes

Reviewer #2: Yes

Reviewer #3: **No: **Unpooled source data for figures is not available. Raw data for insulin secretion experiments is not available. Though transcriptomic and proteomic data have been submitted to public data repositories, I was not able to access the proteomic data (required a password) and it is unclear to me whether only the original RNAseq data is available or processed data too. It would be very helpful to have supplementary tables of these data with lists of proteins or genes and their expression values (as is common in many papers).

PLOS authors have the option to publish the peer review history of their article (what does this mean?). If published, this will include your full peer review and any attached files.

Reviewer #1: No

Reviewer #2: No

Reviewer #3: **Yes: **Michael Ailion and Chau Vuong

---

## [Decision Letter · Decision Letter 1]

21 Mar 2023

Dear Dr Nicholls,

We are pleased to inform you that your manuscript entitled "Insulin secretion deficits in a Prader-Willi syndrome β-cell model are associated with a concerted downregulation of multiple endoplasmic reticulum chaperones" has been editorially accepted for publication in PLOS Genetics. Congratulations!

The revised manuscript was seen by 2 of the original reviewers. (The third reviewer was unavailable and we have decided to proceed with the reviews in hand). As you will see, both reviewers are positive; there are some minor comments that we ask you address during the production process.

Yours sincerely,

Gregory Barsh

Editor-in-Chief

PLOS Genetics

Gregory Copenhaver

Editor-in-Chief

PLOS Genetics

Comments from the reviewers (if applicable):

Reviewer's Responses to Questions

**Comments to the Authors:**

Reviewer #1: I find the authors' response to my concerns to be compelling, and I now believe the manuscript content is appropriate for this journal. No further concerns

Reviewer #2: Most issues have been sufficiently addressed in the updated manuscript or response to reviewers. I still believe that the manuscript would be strengthened by verifying the primary findings in this manuscript (i.e., the chaperone downregulation) in their animal model instead of relying on "indirect evidence" as described in Response #2-4. Nevertheless, the manuscript is suitable for publication without this data. As a minor point, the description of an insulin-deficient mouse model should be clarified on line 533 to note inducible deletion of the Ins2 alleles on the Ins1 global knockout background.

**Have all data underlying the figures and results presented in the manuscript been provided?**

Reviewer #1: Yes

Reviewer #2: Yes

PLOS authors have the option to publish the peer review history of their article (what does this mean?). If published, this will include your full peer review and any attached files.

Reviewer #1: No

Reviewer #2: No

**Data Deposition**

http://datadryad.org/submit?journalID=pgenetics&manu=PGENETICS-D-22-00722R1

**Press Queries**

---

## [Editor Report · Acceptance letter]

13 Apr 2023

PGENETICS-D-22-00722R1 

Insulin secretion deficits in a Prader-Willi syndrome β-cell model are associated with a concerted downregulation of multiple endoplasmic reticulum chaperones 

Dear Dr Nicholls, 

We are pleased to inform you that your manuscript entitled "Insulin secretion deficits in a Prader-Willi syndrome β-cell model are associated with a concerted downregulation of multiple endoplasmic reticulum chaperones" has been formally accepted for publication in PLOS Genetics! Your manuscript is now with our production department and you will be notified of the publication date in due course.

With kind regards,

Anita Estes

PLOS Genetics

On behalf of:
